# Diverse mechanisms of translation arrest by a Clostridia ribosome stalling peptide CliM

Mayu Yoshida[1,8], Felix Gersteuer [2,8], Ole Berendes [3], Keigo Fujiwara [1,4], Haaris A. Safdari [2], Helge Paternoga [2], Hiraku Takada[1,5], Nozomu Obana [6,7], Helmut Grubmüller[3], Lars V. Bock [3], Daniel N. Wilson [2] ✉ & Shinobu Chiba [1] ✉

Ribosome arrest peptides undergo programmed translational stalling in response to changes in the cellular environment to feedback-regulate gene expression. CliM, an arrest peptide in Clostridia, is encoded upstream of the YidC membrane protein insertase gene, but its function and mechanism remain unclear. Here we show that CliM monitors YidC activity to maintain adequate cellular YidC capacity. Interestingly, *Clostridium kluyveri* CliM induces elongation arrest at multiple sense codons, whereas *Clostridioides difficile* CliM causes termination arrest. Cryo-EM-based structural and mutational analyses demonstrate that *C. difficile* CliM adopts multiple α-helices within the nascent polypeptide exit tunnel, where it forms extensive arrest-essential interactions with the ribosome. The residue immediately N-terminal to the stalling site contributes to arrest by sterically interfering with full accommodation of the release factor or aminoacyl-tRNA in the A-site. Molecular dynamics simulations suggest that membrane insertion of CliM induces sequential unwinding of these α-helical structures and relocation of the penultimate residue, thereby triggering arrest release. These findings provide a unified mechanistic framework that explains the distinct arrest behaviors of CliM homologs.

Arrest peptides employ regulated translation arrest to monitor intracellular environments and maintain cellular homeostasis through feedback regulation of gene expression[1–4]. For example, *Escherichia coli* SecM monitors the Sec-dependent protein translocation pathway and regulates expression of the secretion motor ATPase SecA[5,6]. In *Bacillus subtilis*, MifM monitors the YidC-dependent membrane protein insertion pathway and regulates expression of a YidC homolog[7,8]. *Vibrio alginolyticus* VemP also responds to changes in the capacity of the Sec pathway and regulates expression of SecDF2, a component that facilitates the Sec-dependent protein translocation[9]. These arrest peptides are all encoded upstream of their target genes and regulate translation of their downstream genes in cis. In bacteria, protein translocation and membrane insertion mainly occur through the Sec pathway, where the SecYEG complex functions as the protein-conducting channel[10,11]. YidC participates in membrane protein biogenesis both within the Sec pathway and independently as a membrane protein insertase[12,13].

[1]Faculty of Life Sciences and Institute for Protein Dynamics, Kyoto Sangyo University, Kyoto, Japan. [2]Institute for Biochemistry and Molecular Biology, Martin-Luther-King-Platz 6, University of Hamburg, 20146 Hamburg, Germany. [3]Department of Theoretical and Computational Biophysics, Max Planck Institute for Multidisciplinary Sciences, Göttingen, Germany. [4]Department of Gene Function and Phenomics, National Institute of Genetics, Mishima, Japan. [5]Biotechnology Research Center and Department of Biotechnology, Toyama Prefectural University, Toyama, Japan. [6]Transborder Medical Research Center, Institute of Medicine, University of Tsukuba, Tsukuba, Japan. [7]Microbiology Research Center for Sustainability (MiCS), University of Tsukuba, Tsukuba, Japan. [8]These authors contributed equally: Mayu Yoshida, Felix Gersteuer. ✉e-mail: Daniel.wilson@uni-hamburg.de; schiba@cc.kyoto-su.ac.jp

We recently discovered that many bacteria encode unique arrest peptides upstream of *secA*, *secDF*, and *yidC*[14,15]. However, experimental evidence that these peptides actually monitor Sec or YidC activity has not been obtained. Interestingly, many of these arrest peptides contain a conserved arrest motif with a RAPP (Arg-Ala-Pro-Pro) motif or a related sequence[14–16]. In contrast, some arrest peptides lack such motifs and share no detectable sequence similarity with previously described arrest peptides. Among the latter is a factor provisionally designated uC_KYxIW in our previous study[14], which we renamed here "CliM," after **Cl**ostridium **i**nsertase **m**onitor (see below). CliM is encoded upstream of one of the three *yidC* genes of *Clostridium kluyveri* (hereafter referred to as *yidC2*).

The only arrest peptide experimentally shown to monitor YidC activity is MifM, which is encoded upstream of *yidC2*, the secondary YidC homolog[7,8]. In *B. subtilis*, the primary YidC homolog SpoIIIJ is expressed constitutively, while *yidC2* expression is induced via MifM when SpoIIIJ activity decreases[8,17]. The *mifM-yidC2* mRNA forms a stem-loop structure that masks the Shine-Dalgarno (SD) sequence of *yidC2*, thereby repressing its translation. Translation arrest of MifM near its C-terminus stalls a ribosome that keeps the secondary structure unfolded, thereby enabling *yidC2* translation. The arrest is released when the N-terminal transmembrane (TM) segment of MifM is inserted into the membrane in a YidC-dependent manner, ensuring that *yidC2* is expressed only under YidC-limiting conditions, when MifM arrest persists[8]. CliM is also encoded upstream of a *yidC* gene and carries an N-terminal TM segment and a C-terminal arrest motif[14], like MifM, suggesting a similar YidC-monitoring function. To date, however, no experimental evidence has supported this notion. Furthermore, because CliM shows no sequence similarity to known arrest peptides, its arrest mechanism is likely to be distinct, however, this remains to be determined.

In this study, we elucidated the physiological function and molecular mechanism of CliM through genetic, biochemical, structural, and molecular dynamics (MD) simulation analyses. Our results show that CliM monitors and regulates the level of YidC. Structural and mutational analyses revealed that CliM adopts multiple helical conformations within the nascent peptide exit tunnel (NPET) of the ribosome, where its extensive interactions with components of the NPET wall are essential for arrest. The CliM residue immediately N-terminal to the stalling site was found to block full accommodation of the release factor (RF) or aminoacyl-tRNA in the A-site. MD simulations suggest that membrane insertion-dependent arrest release involves sequential unwinding of the helical structures within the NPET. Notably, distinct CliM homologs differ in whether they induce elongation versus termination arrest and whether they stall the ribosome at a single site or multiple sites. The number and positions of stalling sites are flexibly determined by the local sequence context near the peptidyl transferase center (PTC). Together, our results provide a unified mechanistic explanation for the apparently distinct arrest behaviors of different CliM homologs.

## Results

### CliM is a YidC-monitoring arrest peptide

To examine whether *C. kluyveri* CliM monitors the YidC-dependent membrane insertion pathway and regulates the expression of the downstream *yidC2* gene, we conducted in vivo analysis using *B. subtilis*. To monitor expression of the downstream gene, we constructed a *yidC2'-lacZ* reporter in which *lacZ* was fused in-frame after the sixth codon of Ck *yidC2*, with Ck *cliM* placed upstream (Fig. 1a). If CliM monitors the activity of the membrane insertase and regulates downstream gene expression, then inhibition of CliM insertion into the membrane would induce Ck *yidC2*. We introduced either a wild-type reporter (WT) or a reporter lacking residues 12–21 within the transmembrane (TM) region of CliM (ΔTM, Fig. 1a) into the *B. subtilis* chromosome. The reporter assay showed that deletion of the TM

region led to approximately a two-fold increase in β-galactosidase activity compared with WT (260 units vs 467 units) (Fig. 1b, left). Furthermore, the reporter strain carrying the arrest-deficient mutation (K65A/W69A, hereafter referred to as KWm[14]) showed low β-galactosidase activity regardless of the presence or absence of the TM region (94 and 135 units, respectively). These results suggest that inhibition of CliM membrane insertion induces expression of *yidC2* in a CliM arrest-dependent manner.

In a Δ*spoIIIJ* background, where the secondary homolog Bs *yidC2* is induced[8,17], the strain carrying the WT reporter showed higher activity than that in the *spoIIIJ*+ background (348 units), and deletion of the TM region of CliM further increased this activity (487 units; Fig. 1b, right), as previously observed in mutational analyses of MifM[8]. Introduction of the arrest-deficient mutation suppressed the induction regardless of the presence or absence of the TM region (94 and 130 units, respectively). Taken together, these results suggest that inhibition of CliM membrane insertion either by deletion of the TM region or *spoIIIJ* induces Ck *yidC2* through a CliM arrest-dependent mechanism.

To test whether membrane insertion of CliM leads to arrest release, we constructed a *cliM-lacZ* translational fusion reporter (Fig. 1a), in which arrest release would elevate β-galactosidase activity. In the *spoIIIJ*+ strain background, deletion of the TM region of CliM reduced β-galactosidase activity (Fig. 1c, left), indicating that membrane insertion of CliM releases translation arrest. In contrast, introduction of the arrest-deficient mutation resulted in high β-galactosidase activity regardless of the presence or absence of the TM region, as expected. The lower activity of the WT reporter compared with the arrest-deficient mutant suggests that partial arrest persists even with the TM region, likely because *B. subtilis* SpoIIIJ does not efficiently insert heterologously expressed Ck CliM or fails to fully release arrest upon membrane insertion. This notion may also account for the higher *yidC'-lacZ* induction observed for the WT reporter relative to the arrest-deficient mutant (Fig. 1b).

Deletion of *spoIIIJ* further decreased β-galactosidase activity relative to the *spoIIIJ*+ background (Fig. 1c, right), indicating that insertion of CliM by Bs YidC2 is less efficient than by SpoIIIJ. This is consistent with the Ck *yidC2'–lacZ* reporter assay, where *spoIIIJ* deletion partially increased Ck *yidC2'-lacZ* expression (Fig. 1b). In the Δ*spoIIIJ* background, deletion of the CliM TM again reduced β-galactosidase activity, whereas the arrest-deficient mutation increased activity to a similar level in both WT and ΔTM reporters (Fig. 1c, right). Collectively, these results demonstrate that translation arrest of CliM is released upon its membrane insertion.

RNAfold analysis[18] predicted that the Ck *cliM-yidC2* mRNA forms a stem-loop structure encompassing the 3′ end of *cliM* and the translation initiation site of *yidC2* (underlined in Fig. 1a). This structure is expected to mask the translation initiation site of *yidC2* and thereby repress translation, whereas a ribosome stalled near the 3′ end of *cliM* would keep the stem-loop unfolded, enabling *yidC2* expression. To test this, we introduced a mutation that disrupts stem-loop formation (Δstem; Fig. 1a) and examined its effect on *yidC2* expression. Reporter assays showed that, in both *spoIIIJ*+ and Δ*spoIIIJ* backgrounds, *yidC2'-lacZ* was constitutively expressed at a high level when the stem-loop was disrupted, and the effects of the ΔTM and arrest-deficient mutations were largely diminished (Fig. 1d). These results indicate that the stem-loop structure significantly contributes to arrest-dependent regulation of downstream gene expression. Together, our results suggest that CliM functions as an arrest peptide that monitors the activity of the YidC-dependent membrane insertion pathway and regulates downstream gene expression. We therefore designated this arrest peptide as the **Cl**ostridium **i**nsertase **m**onitor (CliM).

### Ck CliM undergoes multi-site stalling

We previously determined the arrest sites of Ck CliM by toeprinting using a *B. subtilis* hybrid PURE system (Bs PURE), in which the *E. coli*

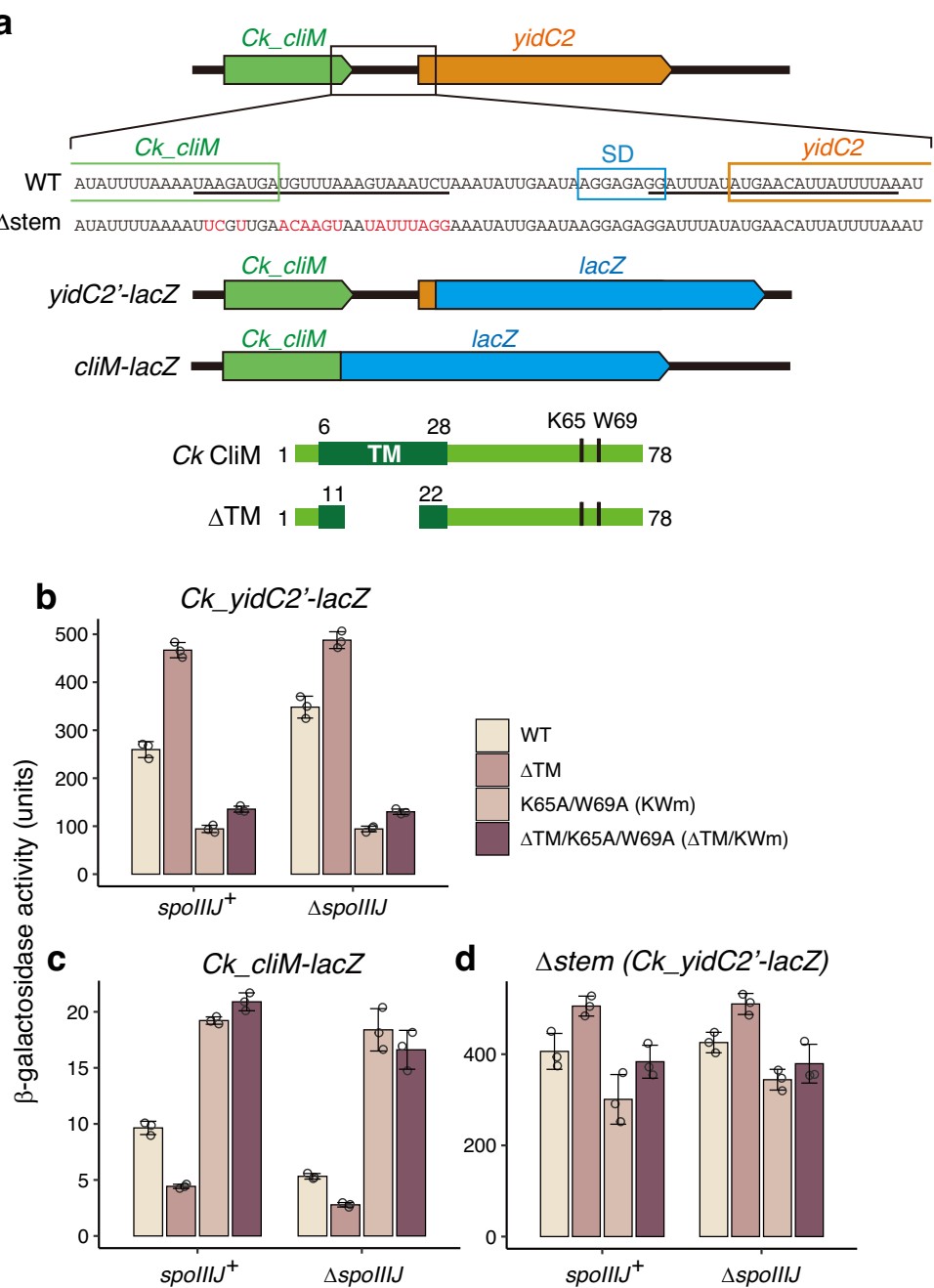

**Fig. 1 | CliM is a YidC-monitoring arrest peptide. a** Schematic representation of the gene context and design of reporter constructs for *C. kluyveri (Ck) cliM* and *yidC2*. mRNA sequences of wild-type (WT) and Δstem mutant derivatives are also shown. Underlines indicate the stem of the predicted stem-loop structure, and red characters represent mutated nucleotides. SD indicates the Shine-Dalgarno sequence. Schematic representations of WT and ΔTM mutant derivatives of CliM with amino acid residue numbers are also shown at the bottom; TM indicates the transmembrane segment. **b–d** β-galactosidase activity (mean ± s.d., *n* = 3, biologically independent cultures) of *B. subtilis* cells harboring WT or mutant derivatives of the downstream gene *lacZ* reporter (*Ck_yidC2'-lacZ*; **b**, **d**) or *Ck_cliM-lacZ* arrest reporter (**c**). ΔTM derivatives carry a deletion of residues 12–21 within the TM region. KWm derivatives carry K65A/W69A substitutions. Δstem mutant derivatives (**d**) carry mutations in the stem-loop region. Source data are provided as a Source Data file.

ribosome in the original PURE system (Ec PURE)[19] is replaced with the *B. subtilis* ribosome[20]. Interestingly, Ck CliM was suggested to cause a major stalling event when the ribosomal P-site reached the Phe75 codon, and a minor stalling event when it reached the Tyr74 codon[14]. To further characterize this stalling behavior, we constructed a hybrid PURE system with ribosomes from *Clostridioides difficile* (Cd PURE), which is more closely related to *C. kluyveri*.

A sandwich-fusion reporter, *gfp-Ck cliM(29–78)-myc-lacZα*, was constructed by fusing a gene segment for the C-terminal region of CliM (residues 29–78) in-frame between *gfp* and *myc-lacZα*. Wild-type and mutant derivatives of this reporter were translated in Bs PURE, Cd PURE, and Ec PURE, and the products analyzed by anti-GFP Western blotting (Supplementary Fig. 1, a–c). As observed previously[14], Bs PURE produced a ~50 kDa band (Supplementary Fig. 1a, lane 1), which shifted to ~30 kDa after RNase treatment (lane 2), consistent with a peptidyl-tRNA arrested at the C-terminal region of Ck CliM. In contrast, the arrest-deficient KWm derivative predominantly accumulated as a full-length, RNase-resistant product (~37 kDa). Similar results were obtained with Cd PURE (Supplementary Fig. 1b), whereas in Ec PURE, both WT and KWm accumulated as full-

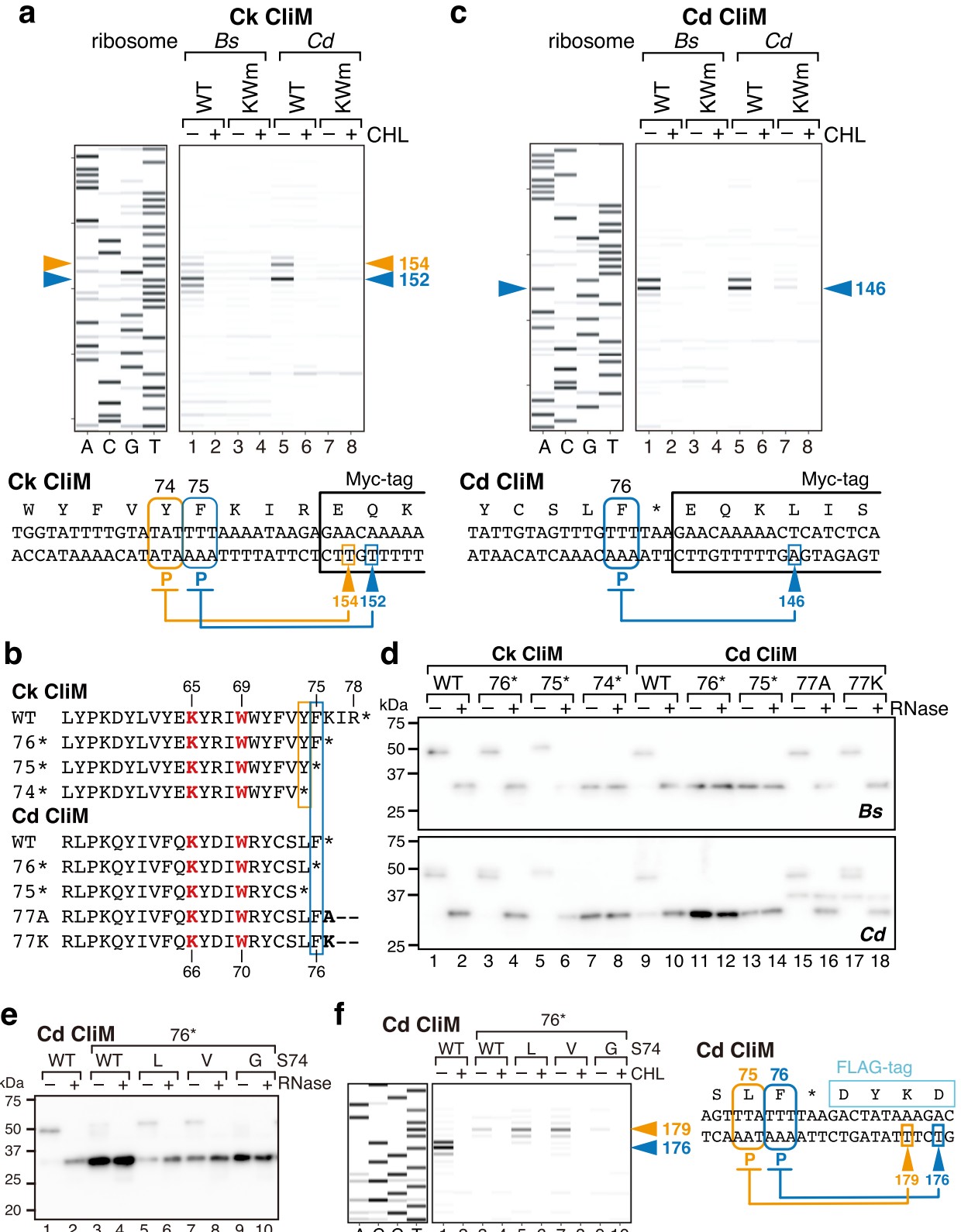

length products (~37 kDa), as reported previously[14] (Supplementary Fig. 1c).

We then performed toeprinting using Bs and Cd PURE to determine the arrest sites. With Bs PURE, multiple toeprint signals were detected, including two distinct peaks at 152 and 154 nt, consistent with previous observations[14] (Fig. 2a, lane 1). These peaks disappeared in the presence of chloramphenicol (CHL; lane 2) or in the KWm (lanes 3, 4). Toeprinting with Cd PURE also revealed multiple signals, including the same two major peaks (Fig. 2a, lanes 5–8). Based on the toeprint sizes, the 152 nt peak corresponded to ribosomes stalled with the P-site at codon 75, while the 154 nt peak corresponded to stalling at codon 74 (Fig. 2a, bottom). Thus, Ck CliM undergoes multi-site stalling at both −1 (codon 74) and 0 (codon 75) sites, and this property is conserved across ribosomes from different species.

**Fig. 2 | Distinct stalling modes between Ck and Cd CliM homologs. a** Toeprinting analysis of Ck CliM in the Bs and Cd PURE. The reverse translation products were analyzed by a capillary sequencer and signals were represented as gel-style heat-maps. In vitro translation was performed in the presence or absence of chloramphenicol (CHL). The toeprint length (nt) was calibrated against dideoxy sequencing (left). The estimated stalling sites (P-site codon) based on the toeprint length are shown with their codon numbers (bottom). **b** Amino acid sequences of WT and mutant derivatives of Ck and Cd CliM used in the experiments shown in d. Mutated residues in KWm mutant are shown in red. Major and minor stalling sites identified in a and b are shown in blue and orange boxes, respectively. **c** Toeprinting analysis of Cd CliM in the Bs and Cd PURE. **d** Western blot analysis of stop codon mutants of Ck and Cd CliM reporters. WT or mutant derivatives of the *gfp-cliM-myc-lacZα* reporters of Ck CliM (lanes 1–8) or Cd CliM (lanes 9–18) were translated in the Bs (upper) and Cd (lower) PURE, and products analyzed by anti-GFP immunoblotting. Samples treated with RNase A (+) were analyzed alongside untreated samples (−) to distinguish peptidyl-tRNA from full-length hydrolyzed products. **e** Western blot analysis of Cd CliM with mutation at Ser74 with a stop codon at the position 76. WT or mutant derivatives were translated in the Bs PURE, and products analyzed by anti-GFP immunoblotting. **f** Toeprinting analysis of WT and Ser74 mutant derivatives of Cd CliM with a premature stop codon at position 76 (76*). All experiments were independently repeated at least twice to ensure reproducibility. Source data are provided as a Source Data file.

## Cd CliM undergoes single-site stalling

To determine whether multi-site stalling is a common feature among CliM homologs, we also analyzed *Clostridioides difficile* CliM (Cd CliM). In Cd CliM, Phe76, corresponding to the major (0-site) stalling position of Ck CliM, is directly followed by a stop codon (Fig. 2b, Source Data 2). Considering the possibility that this stop codon contributes to arrest, we constructed a reporter in which the C-terminal region of Cd CliM (residues 30–76) together with the downstream stop codon was fused between *gfp* and *myc-lacZα* (*gfp-Cd cliM(30–76)*-*myc-lacZα*; * indicates the stop codon). Because of the stop codon immediately before *myc-lacZα*, the translation product should lack the *myc-lacZα* moiety.

Like Ck *cliM*, in vitro translation of Cd *cliM* produced an ~50 kDa RNase-sensitive peptidyl-tRNA product in Bs PURE or Cd PURE, whereas in Ec PURE it produced an ~30 kDa hydrolyzed product lacking tRNA (Supplementary Fig. 1, a–c, lanes 5, 6). In the arrest-deficient KWm derivative (K66A/W70A; Fig. 2b), the product accumulated exclusively as a hydrolyzed form in all systems (Supplementary Fig. 1a–c, lanes 7, 8). These results demonstrate that Cd CliM, like Ck CliM, induces strong translation arrest on *B. subtilis* and *C. difficile* ribosomes.

Toeprinting with Bs PURE and Cd PURE revealed a single peak (Fig. 2c), which disappeared in the presence of chloramphenicol or in the KWm derivative. Based on its size (146 nt), this signal corresponds to stalling at the 76th Phe codon, equivalent to the major (0-site) stalling site of Ck CliM. Because the 77th stop codon is located in the A-site, Cd CliM is suggested to arrest translation during termination.

Together, these findings highlight two key differences between the homologs. First, whereas Ck CliM arrests translation during elongation, Cd CliM arrests during termination. Second, whereas Ck CliM induces multi-site stalling at consecutive codons, Cd CliM stalls at a single site.

## CliM has the potential to arrest both translation elongation and termination

To examine whether Ck CliM possesses the potential to induce termination arrest, we performed stop-codon scanning mutagenesis. Reporter variants carrying stop codons at positions 76 or 75 (76*, 75*; Fig. 2b) predominantly accumulated as RNase-sensitive peptidyl-tRNA products similar to the wild type in both Bs and Cd PURE (Fig. 2d, upper and lower panels, lanes 1–6). Results obtained with the 75* variant indicate that −1 site stalling occurs when a stop codon occupies the A-site, suggesting that Ck CliM can arrest both elongation and termination. Replacement of codon 74 with a stop codon (74*) yielded exclusively hydrolyzed products, suggesting that stalling does not occur at codon 73, in agreement with toeprinting data showing no evidence for −2-site stalling. In the case of Cd CliM, introducing stop codons at positions 76 or 75 (76*, 75*; Fig. 2b) led to accumulation of hydrolyzed products (Fig. 2d, lanes 9–14), consistent with toeprinting results indicating that Cd CliM arrests exclusively at codon 76 (Fig. 2c).

To test whether Cd CliM can also induce elongation arrest, we replaced the native stop codon at position 77 with sense codons (Ala or Lys; 77A, 77K; Fig. 2b) and analyzed the translation products by Western blotting. In Bs PURE, these variants mainly accumulated as peptidyl-tRNA products (Fig. 2d, lanes 15–18, upper panel). In Cd PURE, full-length products were also detected at low levels, but peptidyl-tRNA species remained the major products (lower panel). These results indicate that Cd CliM also has the potential to induce elongation arrest. Collectively, these findings suggest that both Ck CliM and Cd CliM possess the ability to arrest translation during either elongation or termination.

## Latent potential of Cd CliM for −1-site stalling

We hypothesized that the lack of −1-site stalling in Cd CliM was due to the difference in the amino acid sequence context immediately upstream of the stalling site between Cd and Ck CliM. To test whether the identity of the residue N-terminal to the −1-stalling site (Val73 in Ck CliM and Ser74 in Cd CliM) accounts for this difference, we employed the 76* variant of Cd CliM, which predominantly accumulated as a hydrolyzed species due to the absence of −1-site stalling (Fig. 2d, lanes 11, 12). Western blotting followed by in vitro translation with Bs PURE revealed that substitution of Ser74 with Leu or Val significantly increased the proportion of peptidyl-tRNA species (Fig. 2e, lanes 5–8), compared with the Ser74 variant (WT; lanes 3, 4), indicating the crucial role of residue 74 in −1-site stalling. Toeprinting of native Cd CliM yielded 0-site stalling signal (176 nt under these conditions; Fig. 2f, lanes 1, 2), whereas the 76* variant with Ser74 showed no 0-site and only a weak −1-site stalling signal (179 nt; lanes 3, 4). Substitution of Ser74 with Leu or Val significantly enhanced the −1-site toeprint signal (lanes 5–8). Conversely, replacing Ser74 with Gly abolished both peptidyl-tRNA accumulation and toeprint signals (Fig. 2e,f, lanes 9, 10). Together, these results demonstrate that Cd CliM, like Ck CliM, possesses the potential to induce −1-site stalling, which remains latent in the native sequence context due to the presence of Ser74.

## Identification of critical residues in Cd CliM by deep mutational scanning

To comprehensively identify residues critical for translation arrest by Cd CliM, we performed deep mutational scanning (DMS)[21]. A reporter was constructed in which the coding region for the C-terminal soluble domain of Cd CliM (residues 30–76) was fused in-frame between *gfp* and the spectinomycin resistance gene (*spcR*) (Fig. 3a). The native stop codon of Cd CliM was replaced with Lys to generate a continuous *gfp-Cd cliM-spcR* translational fusion. We then generated a plasmid library in which each codon from residues 38 to 76, as well as the introduced Lys77 codon, was individually substituted with a single amino acid. The library was integrated into the *B. subtilis* chromosome. In this system, loss of arrest allows readthrough into *spcR*, conferring spectinomycin (Spc) resistance. Cells were grown to mid-log phase in Spc-free medium, diluted 10-fold into medium with or without Spc, and further cultured to mid-log phase. The cells were then harvested, and the frequencies of individual mutations were quantified by next-generation sequencing (Fig. 3b). Fitness values in the presence of Spc were calculated and visualized as a heatmap (Fig. 3c, Supplementary Fig. 2, and Source Data 2). As expected, the wild-type reporter displayed relatively low fitness, reflecting stable arrest upstream of *spcR*. In contrast, numerous mutants showed increased Spc resistance

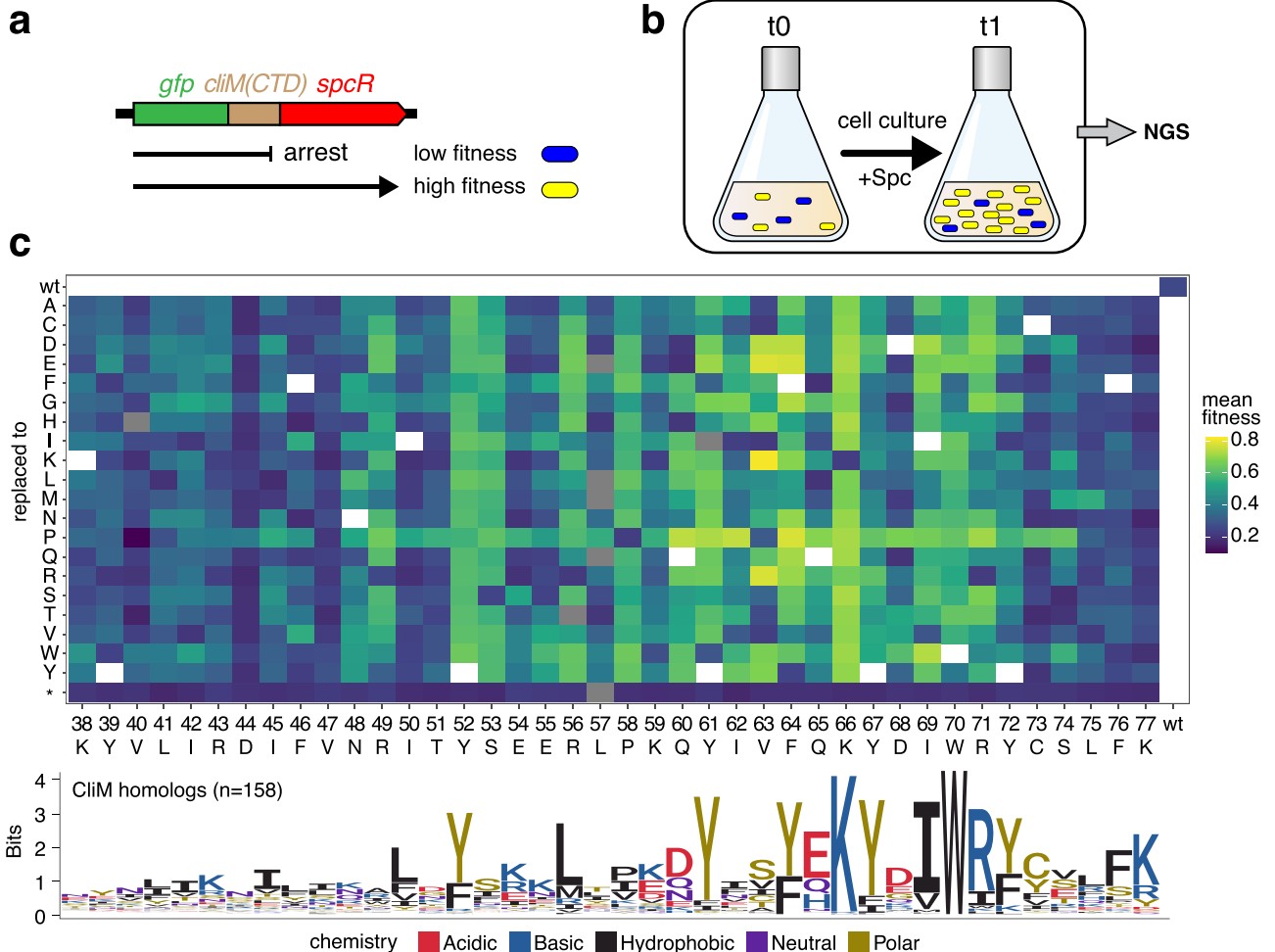

**Fig. 3 | Deep mutational scanning identifies crucial residues of Cd CliM.**
**a** Schematic representation of *gfp-Cd_cliM-spcR* translational fusion reporter. A gene fragment encoding the C-terminal domain (CTD) of Cd CliM was fused in-frame with *gfp* and *spcR*. Loss of arrest allows translation of *spcR*, conferring higher fitness in Spc-containing medium. **b** Workflow of DMS. A library of *B. subtilis* cells carrying the reporter with single point mutations in *cliM* was cultured in the presence of Spc. Fitness was estimated by NGS-based quantification of clone frequencies. **c** Heatmap of relative fitness (mean of two biological replicate) of each mutant. Residue numbers and WT amino acids are indicated at the bottom; substituted residues are shown on the left. A sequence logo depicting amino acid conservation among CliM homologs is shown below the heatmap. Source data are provided as a Source Data 2 file.

(yellow in the heatmap). Notably, substitutions at residues R49, Y52, S53, R56, P58, Q60, Y61, F64, K66, I69, W70, and R71 frequently enhanced Spc resistance, suggesting that the unique properties of these residues are critical for translation arrest. In addition, most substitutions to proline at residues 48–74 increased Spc resistance, possibly indicating that an α-helical structure in this region may contribute to arrest stability. Interestingly, substitutions at residue 76, which corresponds to the stalling site, and adjacent positions had little impact on Spc resistance.

## Cryo-EM structures of CliM-stalled ribosomal complexes

To provide insight into the mechanism by which CliM induces translational arrest, we set out to determine the structure of a CliM-stalled ribosomal complex using single particle cryo-EM. Since CliM was shown to stall efficiently on *B. subtilis*, but not *E. coli*, 70S ribosomes (Supplementary Fig. 1a, c), we employed a *B. subtilis* cell-free in vitro translation extract to generate CliM-stalled ribosomal complexes (CliM-SRC), as used previously to generate *B. subtilis* MifM- and, ApdA-SRCs[16,22]. We employed the Cd rather than Ck CliM sequence since our toeprinting assays showed that Cd CliM stalls at single, defined site with the Phe76 codon in the P-site and a stop codon in the A-site (Fig. 2c). By contrast, Ck CliM stalls at multiple sites (Fig. 2a), which

would increase heterogeneity and may be computationally hard to distinguish. Similar to the ApdA- and ApdP-SRCs[16], the CliM-SRC was purified using an N-terminal FLAG-tag exposed at the exit tunnel and was formed with a full-length, rather than truncated, peptide coding sequence (see "Methods"). The CliM-SRC was applied to cryo-grids and a total of 11,409 micrographs were collected on a Titan Krios transmission electron microscope equipped with a K3 direct electron detector, yielding 854,071 ribosomal particles after 2D classification (Supplementary Fig. 3). After multiple rounds of 3D classification, three defined subpopulations of 70S ribosomes emerged; two major states with tRNA in the P-site, and either a vacant A-site (43%; 368,433 particles) or with RF bound in the A-site (41%, 348,223), as well as one minor state bearing A- and P-site tRNAs (6%, 47,146 particles) (Supplementary Fig. 3). These three states were refined to average resolutions of 2.3, 2.3, and 2.8 Å, respectively (Supplementary Fig. 3, 4 and Table 1).

In the CliM-SRC with RF bound in the A-site (Fig. 4a), the CliM nascent polypeptide chain (NC) spanning from the PTC to the vestibule of the NPET was well-resolved, enabling 38 amino acids (Tyr39-Phe76), including sidechains, to be modeled unambiguously (Fig. 4b, Supplementary Fig. 5 and Supplementary Movie 1). Similarly, in the CliM-SRC with the vacant A-site or presence of A-tRNA, the CliM NC

**Table 1 | Cryo-EM data collection, refinement and validation statistics**

| Model | CliM SRC with P-tRNA and RF | CliM SRC with P-tRNA and vacant A-site | CliM SRC with P- and A-tRNA |
|---|---|---|---|
| EMDB ID | EMD-50855 | EMD-50856 | EMD-50858 |
| PDB ID | 9FY1 | 9FY2 | 9FY3 |
| Data collection and processing | | | |
| Magnification (×) | 105,000 | 105,000 | 105,000 |
| Electron fluence ($e^-/Å^2$) | 40 | 40 | 40 |
| Defocus range (μm) | −0.3 to −0.9 | −0.3 to −0.9 | −0.3 to −0.9 |
| Pixel size (Å) | 0.831 | 0.831 | 0.831 |
| Initial particles | 854,071 | 854,071 | 854,071 |
| Final particles | 348,223 | 368,433 | 47,146 |
| Average resolution (Å) (FSC threshold 0.143) | 2.3 | 2.3 | 2.8 |
| Model composition | | | |
| Atoms | 138,649 | 137,122 | 139,158 |
| Protein residues | 5579 | 5341 | 5340 |
| RNA bases | 4435 | 4432 | 4528 |
| Refinement | | | |
| Map CC around atoms | 0.93 | 0.93 | 0.92 |
| Map CC whole unit cell | 0.89 | 0.89 | 0.88 |
| Map sharpening B factor ($Å^2$) | −52.75 | −52.13 | −50.59 |
| R.M.S. deviations | | | |
| Bond lengths (Å) | 0.009 | 0.008 | 0.009 |
| Bond angles (°) | 1.886 | 1.814 | 2.019 |
| Validation | | | |
| MolProbity score | 1.23 | 1.13 | 1.73 |
| Clash score | 1.19 | 1.13 | 1.65 |
| Poor rotamers (%) | 1.07 | 1.34 | 2.59 |
| Ramachandran statistics | | | |
| Favored (%) | 94.52 | 94.77 | 91.65 |
| Outlier (%) | 0.33 | 0.31 | 0.76 |
| Ramachandran Z-score | −2.82 | −2.52 | −4.01 |

was less well-resolved, however, the conformation and majority of sidechains were consistent with that observed in the presence of RF (Supplementary Fig. 5). Like the CliM-SRC with RF, the density for the P-site tRNA is consistent with the presence of tRNA$^{Phe}$, suggesting that all three states are arrested with the Phe76 codon in the P-site (Supplementary Fig. 6a–c) and with the UAA stop codon in the A-site. For the A-tRNA bound state, we propose that tRNA$^{Tyr}$, which normally decodes UAC and UAU codons, has miscoded the UAA codon (Supplementary Fig. 6d–f), however, the lower resolution of this complex makes an unequivocal assignment difficult. Interestingly, in this state, the CCA-end and Tyr moiety of the A-tRNA$^{Tyr}$ are poorly resolved (Supplementary Fig. 6g–i), indicating that although the tRNA$^{Tyr}$ has accommodated on the 50S, the CCA-end is not stably accommodated at the PTC.

## The CliM arrest peptide adopts a compacted conformation in the NPET

In the CliM-SRC, residues Leu47–Cys73 of the CliM NC are highly compacted, forming α-helical or α-helical-like conformations (Fig. 4c,d), with the outcome that these 27 residues, which could almost span the entire NPET (94.5 Å) in an extended conformation (3.5 Å/residue), span only 37 Å−significantly less than half of the NPET. The overall conformation and path of the CliM NC is reminiscent of that of other arrests peptide, such as SecM[23], TnaC[24,25], hCMV[26], but especially VemP[27] (Supplementary Fig. 7). However, unlike VemP, where α-helical formation occurs directly at the PTC, the C-terminal residues of CliM, like SecM and TnaC, are not compacted (Fig. 4c,d and Supplementary Fig. 7). Because of the overall compacted nature of the CliM NC, the residues of the $_{66}$KYxIW$_{70}$ motif of CliM are not located deep in the NPET, but are rather positioned adjacent to the PTC, directly before the constriction created by ribosomal proteins uL4 and uL22 (Fig. 4c). The α-helical nature of CliM is illustrated by the 14 intramolecular hydrogen bonds that can be formed between the carbonyl oxygen of one amino acid and the amide hydrogen of another amino acid four residues down the chain (i + 4). In addition, salt bridges are possible with the backbone from sidechains of Ser74, Lys66, Gln 60, Lys59, Arg56, Ser53, Thr51, and Asn48, and between the sidechains of Arg71 and Asp68 as well as Gln60 and Arg56 (Fig. 4d). The importance of the α-helical conformation is supported by the observation that most proline substitutions within residues 48–74 led to increased Spc resistance (Fig. 3c and Supplementary Fig. 8), indicating a reduction in the stalling activities of these CliM variants.

## Interaction of CliM with 23S rRNA components of the NPET

The CliM NC establishes extensive contacts with the 23S rRNA components of the NPET (Fig. 5a–h, Supplementary Fig. 9). This includes five stacking interactions between sidechains of CliM residues and nucleotides of the 23S rRNA, including Tyr72 with C2610 (BsC2639), Arg71 and U2586 (BsC2615), Trp70 and A2062 (BsA2091), Arg56 together with Gln60 associated by salt bridges and A751 (BsA798), as well as Tyr52 and A1614 (BsA1659) (Figs. 5a–c, g, h). Consistently, our mutagenesis analysis also indicated that Arg71, Trp70, Arg56 and Tyr52 are important for CliM-dependent stalling activity (Fig. 3c). In addition, seven sidechains of CliM can hydrogen bond with 23S rRNA nucleotides, Ser74 with A2503 (BsA2532), Tyr67 with G748 (BsG795), Lys66 with A751 (BsA798), Gln65 with A2062 (BsA2091), Tyr52 with A751 (BsA798), and A1614 (BsA1659), Asn48 with C461 (BsC508) (Fig. 5e, h, Supplementary Fig. 9). Of those, mutation at Lys66 and Tyr52 led to reduced CliM arrest activity (Fig. 3c). Indeed, Lys66, together with Trp70, represents the highly conserved K and W of the KYxIW motif of CliM, and mutation of K66A and W70A generates the KWm variant that was shown to be inactive in vivo (Fig. 1) and in vitro (Supplementary Fig. 1a–c) for arrest activity. We also predict three additional hydrogen bonds between the backbone of CliM and 23S rRNA nucleotides, namely, Tyr72 with G2505 (BsG2534), Tyr67 with U2609 (BsU2638), and Lys59 with U790 (BsU837) (Supplementary Fig. 9). Ile69 and Phe64 that were shown by mutagenesis to be important for CliM-mediated translational arrest (Fig. 3c), form van der Waals and hydrophobic interactions by inserting into clefts formed by 23S rRNA nucleotides C2610/C2611 (BsC2639/C2640) and A2058/A2059/C2611 (BsA2087/A2088/C2640), respectively (Fig. 5d, f).

## Interaction with uL22 is critical for CliM-mediated translational stalling

In addition to contacts with the nucleotides of the 23S rRNA, the CliM nascent chain makes extensive interaction with β-hairpin of uL22 (Fig. 6a). At the tip of uL22, the sidechain of Arg92 is stacked upon by the sidechain of Tyr61 of CliM (Fig. 6b). This interaction is likely to be critical for stalling since mutation of Tyr61 with most other amino acids relieved stalling (Fig. 3c). The exception was mutations to other aromatic amino acids, such as Phe or Trp that maintained arrest activity (Fig. 3c), presumably because they are able to form similar stacking interactions. The sidechain of Arg92 of uL22 also comes within hydrogen bonding distance to the backbone of Ser53 (Fig. 6b), another residue shown to be critical for stalling (Fig. 3c). Conversely, the

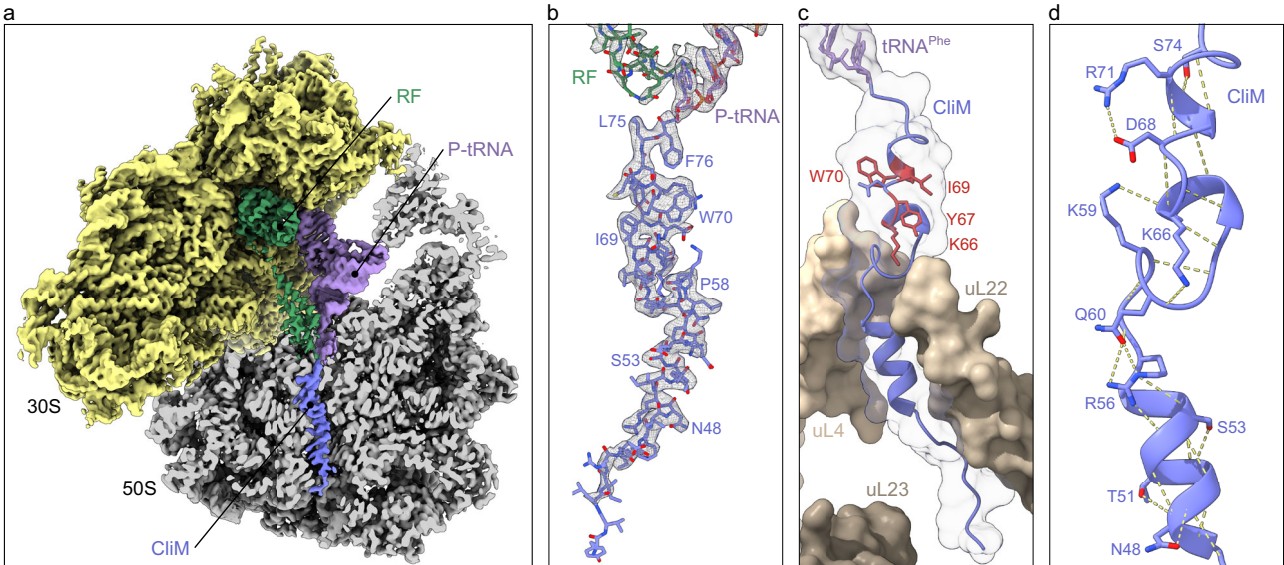

**Fig. 4 | Overview of CliM-SRC with RF from cryo-EM. a** Cryo-EM map of the 3D-refined *B. subtilis* CliM-SRC with transverse section of 50S (gray) to reveal density for the nascent chain (blue), P-tRNA (lavender), release factor (green) and 30S (yellow). **b** Cryo-EM map density (black mesh) for P-site tRNA with attached nascent chain and release factor of the 3D-refined *B. subtilis* CliM-SRC. The P-tRNA (lavender) bears the CliM nascent chain (blue), and the A-site is occupied by the release factor (green). **c** P-tRNA (lavender) and CliM (blue) in relation to uL4 (light gold), uL22 (gold) and uL23 (dark gold) in surface representation, with residues of conserved KYxIW motif shown in red. **d** Intermolecular contacts (dashed yellow lines) of CliM (blue).

backbone of Phe85 of uL22 can hydrogen bond with the sidechain of Arg49 of CliM (Fig. 6c), which was also shown by mutagenesis to be important for arrest activity (Fig. 3c). In addition, hydrogen bonds are also possible between the backbone of Arg43 and Ile42 of CliM with Lys83 and Arg84 of uL22, respectively (Fig. 6d, e). By contrast, we observe no contact of CliM with uL23 (Fig. 6a), and the closest point of contact with uL4 is with Pro58, which is 3.5–3.6 Å from Gly69 and Arg66 located at the tip of uL4 (Fig. 6f).

To assess the importance of uL4, uL22 and uL23 for CliM-mediated translational arrest, we performed in vivo reporter assays in *B. subtilis*. For Ck CliM, we used the reporter *gfp–Ck cliM(29–78)–lacZ*, and for Cd CliM, we used *gfp–Cd cliM(30–76)–77K–lacZ*, in which the native stop codon at position 77 of Cd CliM was replaced with Lys (Fig. 6g and Supplementary Fig. 10a). Each reporter and its arrest-defective variant (KWm) were integrated into the *B. subtilis* chromosome, and β-galactosidase activity was measured. As expected, the wild-type reporters exhibited low activity, whereas the KWm variants showed high activity (Fig. 6h). We next introduced these reporters into *B. subtilis* strains carrying 5-residue deletions in the loops of ribosomal proteins uL4, uL22, or uL23 that protrude into the NPET (Fig. 6i–k) (uL4_Δ66–70, uL22_Δ86–90, uL23_Δ65–69; shown in red in Fig. 6a). In both Ck and Cd CliM reporters, the uL22 mutant strain showed increased β-galactosidase activity to a level comparable with the KWm (Fig. 6j and Supplementary Fig. 10b). By contrast, the uL4 and uL23 mutant strains retained low activity similar to the wildtype (Fig. 6i, k and Supplementary Fig. 10c, d). These results suggest that interaction between CliM and uL22 indeed plays a critical role in translation arrest, consistent with the strong interaction observed between CliM and uL22, but not uL4 or uL23 (Fig. 6a–e).

**CliM stalls translation by occluding the A-site of the PTC**

Our biochemical analysis demonstrated that ribosomes can translate Cd CliM until Phe76, when the stop codon enters the A-site, but cannot undergo peptidyl-tRNA hydrolysis to release the nascent chain (Fig. 2). This does not appear to be due to lack of binding of the release factors since we observe that 41% of the particles contain a release factor bound in the A-site (Supplementary Fig. 3). To understand why peptidyl-tRNA hydrolysis has not occurred, we compared the position

of RF determined here in the context of the CliM-SRC with the position of RF1 trapped on the ribosome after accommodation at the PTC but directly before peptidyl-tRNA hydrolysis (Fig. 7a)[28]. This comparison illustrates that the overall conformation of RF in the CliM-SRC is very similar to that determined previously, having adopted an open conformation with domain III of the RF extending into the PTC (Fig. 7a). However, in the CliM-SRC, the GGQ-loop, located at the tip of domain III, is displaced away from the PTC by 3–4 Å (Fig. 7a–c and Supplementary Movie 1). In fact, the accommodated position of the GGQ-loop in the PTC of the CliM-SRC is not possible due to steric clashes with Leu75 of the CliM nascent chain (Fig. 7c and Supplementary Movie 1). Thus, it appears that CliM-mediated translational stalling arises because Leu75 encroaches on the A-site of the PTC and thereby prevents full accommodation of the GGQ-loop of RF.

In silico mutagenesis of Leu75 to smaller amino acids, such as Gly, would be predicted to reduce the overlap with the GGQ-loop of RF1 (Fig. 7d) and, therefore, might be expected to reduce the stalling efficiency, however, this prediction contradicted our mutagenesis results (Fig. 3c). To resolve this discrepancy, we analyzed L75A and L75G variants by toeprinting. Because we used a construct in which the stop codon of Cd CliM was replaced with Lys (77 K) in DMS, the same context was employed for toeprinting in Bs PURE. As expected, a major toeprint signal at 176 nt was observed (Supplementary Fig. 11, lanes 1,2; WT/77 K), corresponding to stalling at Phe76 (0-site stalling). This signal was abolished in the KWm variant (Supplementary Fig. 11, lanes 7,8; KWm/77 K). Then, we replaced Leu75 with smaller residues (L75A or L75G) in the 77 K context. Strikingly, in both cases, the 176 nt toeprint signal disappeared, and a new signal emerged at 173 nt (Supplementary Fig. 11, lanes 3–6; L75A/77 K, L75G/77 K). This result indicates that the L75A and L75G substitutions abolished 0-site stalling and instead induced stalling one codon downstream, at Lys77 (+1-site stalling).

Importantly, in the wild-type context with a stop codon at position 77 (77*), the 176-nt toeprint signal observed for the wild-type was markedly reduced or eliminated when L75 was replaced with Ala, Gly, or Ser (Fig. 7e, lanes 1–8). Western blotting further revealed that, while the wild-type accumulated predominantly as peptidyl-tRNA, the L75A mutant exhibited decreased peptidyl-tRNA levels accompanied by

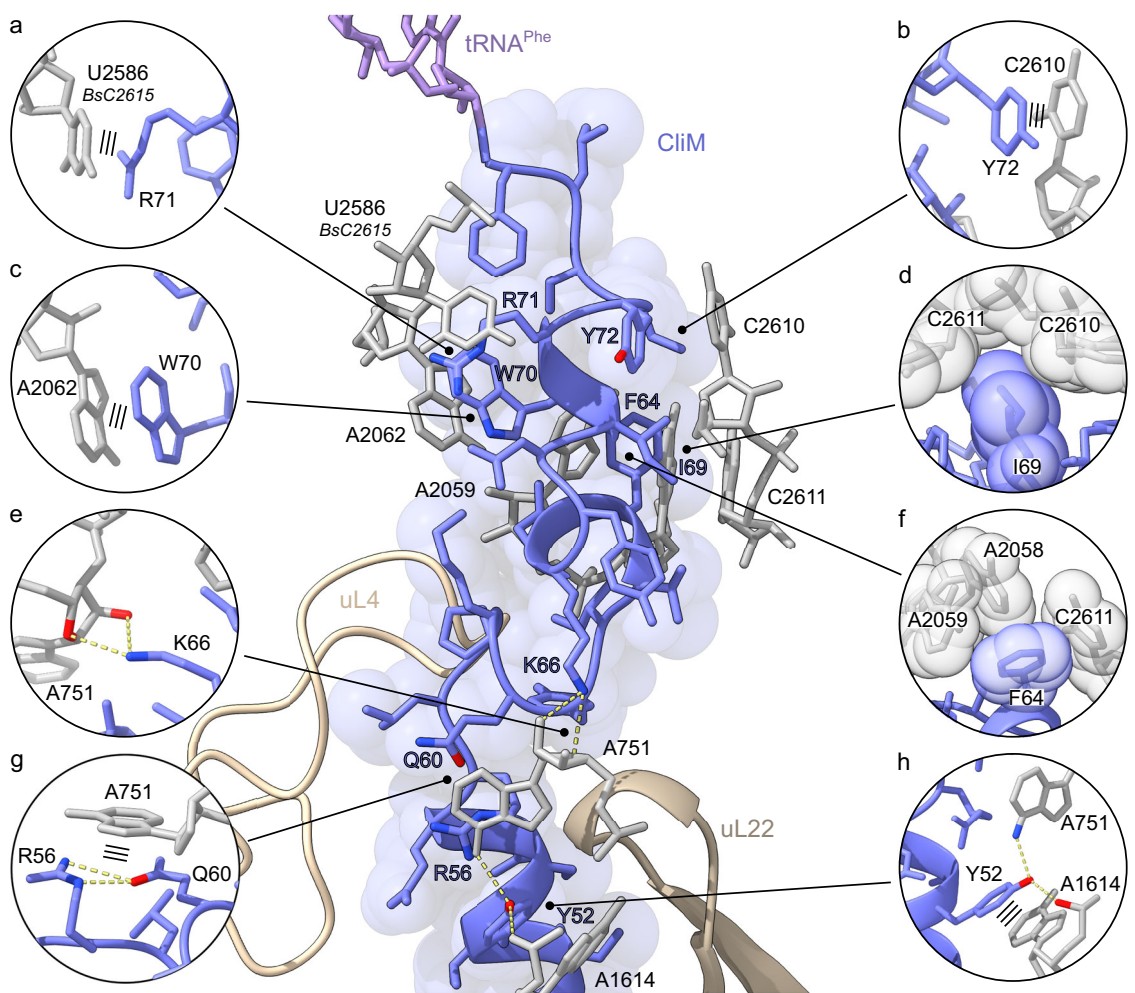

**Fig. 5 | Interaction of CliM with nucleotides of the 23S rRNA. a–h** The central panel shows the CliM nascent chain (blue) attached to the P-site tRNA (lavender) and ribosomal proteins uL4 (light gold) and uL22 (gold), with selected contacts with the 23S rRNA (gray) highlighted by individual panels. Dashed yellow lines indicated potential hydrogen bond interactions, whereas three black lines indicate probable stacking interactions.

increased levels of hydrolyzed products (Fig. 7f, lanes 1–4). Substitution of L75 with Gly or Ser almost completely abolished detectable peptidyl-tRNA (Fig. 7f, lanes 5–8). By contrast, substitutions with larger residues, such as Ile, Val, or Trp, preserved both the toeprint signal and peptidyl-tRNA accumulation (Fig. 7e, f). These results indicate that residues with a small sidechain at position 75 cannot maintain stable 0-site stalling. This is consistent with structural data showing a steric clash between the Leu75 sidechain and the RF.

**Molecular dynamics simulations on CliM stalling and relief of arrest**

To study if the conformational dynamics of the Cd CliM peptide at physiological temperatures affects the overlap with the canonical position of the RF, we performed all-atom molecular dynamics (MD) simulations of CliM within the NPET. We additionally carried out simulations of the L75A and L75G variants of CliM. During the simulations, the overlap volume between WT CliM and the aligned accommodated RF (PDB ID 4V5J)[29] was always larger than 23 Å³ (Fig. 8a), while for the L75A variant, markedly smaller overlap volumes were observed (reaching 10 Å³). In the case of L75G, we observed multiple peptide conformations that did not overlap with RF. These results support the notion that the large L75 sidechain sterically prevents the RF from obtaining a hydrolysis-competent conformation, while the overlap with the smaller alanine sidechain only provides enough hindrance to slow down release, as indicated by the increased

levels of hydrolyzed products (Fig. 7f). The absence of a sidechain in glycine further reduces hindrance and therefore does not result in stalling.

To check if, in addition to side-chain size, also the accessible conformations of the CliM peptide affect the overlap volume, we performed a functional mode analysis[30] to identify the conformational mode of the C-terminal backbone atoms (C73–F76) that best predicts the volume (correlation coefficients 0.85 and 0.64 for training and validation sets, Supplementary Fig. 12a). This functional mode corresponds to a shift of the peptide into the tunnel and a stretching of its backbone. For L75A and L75G variants, lower values of the functional mode are reached, which correspond to lower overlap volumes (Fig. 8a), showing that the conformational flexibility also contributes to the reduced overlap.

As membrane insertion of the TM segment of CliM is likely to result in a pulling force on the CliM peptide and relieve stalling (Fig. 1), we performed pulling simulations to investigate the mechanism by which the pulling force mediates relief. In the pulling simulations, a harmonic spring potential was applied to the N-terminal Tyr39. In the course of the simulations, the spring position was moved with constant velocity by 82 Å in the direction of the tunnel axis, thereby exerting a force on Tyr39. To test the dependence of our results on the pulling velocity, we performed multiple sets of pulling simulations with velocities of 4, 8, 16, 32, 64, and 128 mm/s, each with 8 replicates. Independent of the velocity, the extension of the peptide resulted in a

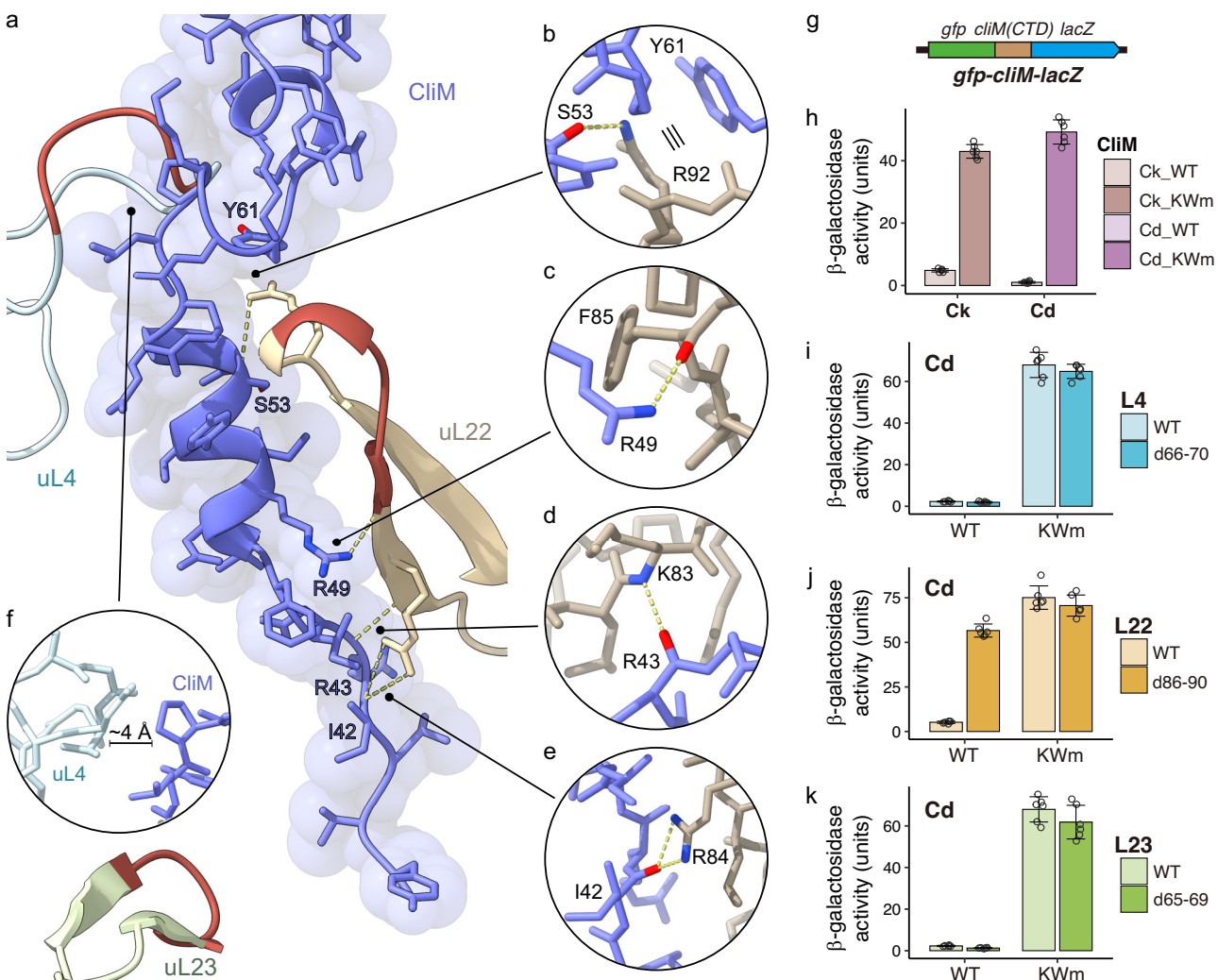

**Fig. 6 | Contacts of CliM with ribosomal protein uL22. a** The central panel shows the CliM nascent chain (blue) and ribosomal proteins uL4 (light blue), uL22 (beige), and uL23 (light green), with **b**–**e** selected interactions between CliM and uL22 highlighted, whereas in contrast, **f** only minimal contact with uL4 is observed. Potential hydrogen bonds are shown as dashed yellow lines and stacking interactions as three black lines. **g** Schematic representation of *gfp-cliM-lacZ* reporter. A gene fragment encoding the C-terminal domain (CTD) of CliM was fused in-frame with *gfp* and *lacZ*. **h** β-galactosidase activity (mean ± s.d., *n* = 6, biologically independent cultures) of *B. subtilis* cells carrying WT or KWm derivatives of the *gfp-cliM-lacZ* reporter. **i**–**k** β-galactosidase activity (mean ± s.d., *n* = 6, biologically independent cultures) of *B. subtilis* cells carrying WT or KWm derivatives of the *gfp-cliM-lacZ* reporter with (dark bars) or without (light bars) loop deletions in uL4 (**i**), uL22 (**j**), or uL23 (**k**). Toeprinting and Western blotting were independently repeated at least twice to ensure reproducibility. Source data are provided as a Source Data file.

sequential unfolding of the CliM helices from the N- to C-terminus (Fig. 8b and Supplementary Fig. 12b). When the helices are fully unfolded, the C-terminal backbone conformation changes along the functional mode identified from the equilibrium simulations (Fig. 8c, left) and the overlap volume drops to zero (Fig. 8c, right), which would allow the RF to fully accommodate and release the CliM peptide. Thus, our MD simulations suggest that relief of CliM-mediated translation stalling requires unfolding of all secondary structure within the CliM NC before the pulling force removes the steric hindrance of Leu75 and allows RF or A-tRNA accommodation.

## Discussion

Our cryo-EM analysis of Cd CliM, together with biochemical and in vivo mutational analyses, elucidated the molecular mechanism underlying ribosome stalling by CliM (Fig. 9a–c). Cd CliM induces termination arrest when the codon for Phe76 resides in the P-site, and the stop codon occupies the A-site. In this state, an RF can bind to the A-site, but full accommodation of the GGQ loop of the RF is hindered by steric interference from the sidechain of Leu75 (Fig. 9a, b). By contrast, when

Leu75 is mutated to Gly in CliM, the steric clash is relieved (Fig. 9c), and hence the translational arrest does not occur, as observed biochemically (Fig. 7e, f) and in MD simulations (Fig. 8a). We propose that the same mechanism of action is used to arrest translation elongation since superimposition of an accommodated A-site tRNA with the CliM NC in the CliM-SRC also leads to steric overlap between Leu75 and the aminoacyl moiety of the accommodated A-site tRNA (Supplementary Fig. 13a–c). Consistently, we observe that the aminoacylated CCA-end of the A-site tRNA is disordered in a small population of the CliM-SRCs where the stop codon has been decoded by an A-site tRNA, which we presume is Tyr-tRNA$^{Tyr}$ (Supplementary Fig. 6). Moreover, based on our Cd CliM-SRC structure and the high sequence similarity between the Ck and Cd CliM arrest motifs, we hypothesize that Ck CliM utilizes a similar mechanism as Cd CliM to stall translation, at least at the major stall site with codons for Phe75 in the P-site and Lys76 in the A-site (Fig. 2a). Specifically, we propose that the Ck CliM adopts a similar structure to Cd CliM and that stalling occurs due to a steric clashing between the sidechain of Tyr74 of Ck CliM, which is equivalent to Leu75 of Cd CliM, and the Lys moiety of the A-site tRNA

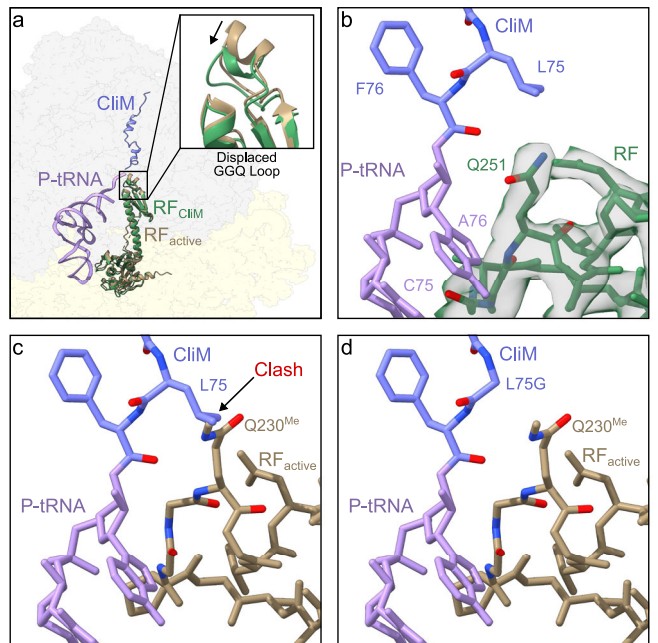

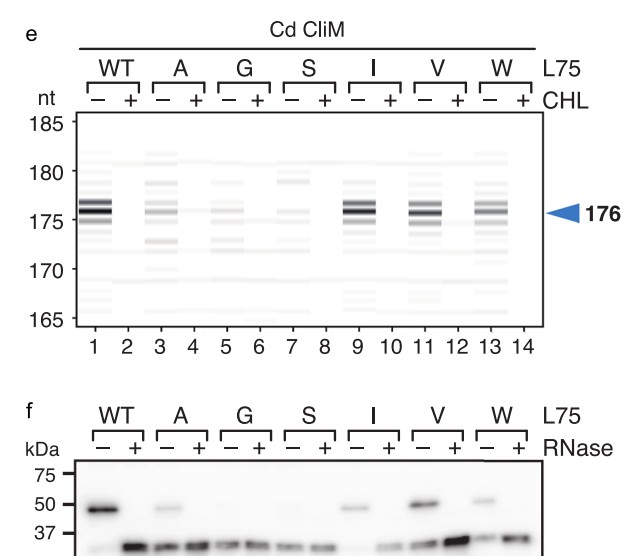

**Fig. 7 | Mechanism for CliM-mediated PTC arrangement leading to translational stalling. a** Overview of the CliM-SRC with RF (green) bound in the A-site and the nascent chain (blue) attached to a tRNA (lavender) at the P-site, overlayed with the position of canonical position of RF (brown) (PBD ID 9MTP)[28]. Inset shows the displacement of the GGQ-loop between the CliM-SRC RF (green) and canonical RF (brown) position. **b** View of the PTC of the CliM-SRC with RF (green model with transparent cryo-EM map density) in the A-site and the nascent chain (blue) attached to a tRNA (lavender) at the P-site. **c** Same view as in (**b**) with a properly accommodated release factor (brown) in the A-site (PDB ID 9MTP)[28] and a tRNA (purple) lacking a nascent chain in the P-site. **d** Same view as in (**c**) but with in silico mutation of Leu75Gly (L75G) in the CliM-SRC, alleviating the steric overlap with the Q230^Me of RF (brown). **e** Toeprinting analysis of WT and Leu75 mutant derivatives of Cd CliM. In vitro translation was performed in the presence or absence of chloramphenicol (CHL). The toeprint length (nt) was calibrated against dideoxy sequencing. **f** Western blot analysis of the in vitro translation products of WT and Leu75 mutant derivatives of the *gfp-cliM* reporters using the Bs PURE. The products were detected using anti-GFP antibody. Samples treated with RNase A (+) were analyzed alongside untreated samples (−). Source data are provided as a Source Data file.

(Supplementary Fig. 13d–f). Thus, our model explains why Cd and Ck CliM can induce both termination and elongation arrest, namely, due to blockage of either the GGQ loop of the RF or the aminoacyl-moiety of the accommodating A-tRNA at the A-site, respectively.

Such a mechanism to inhibit accommodation at the A-site is shared by several other arrest peptides, including CMV, TnaC, MifM and VemP[22,24,25,27,31], however, in these cases, conformational changes in 23S rRNA nucleotides occur to sterically block the accommodation at the PTC. By contrast, in the CliM-SRC, it is exclusively the sidechain at the penultimate position of the CliM NC that generates the overlap with the RF or aminoacyl-moiety of the A-tRNA (Fig. 7c and Supplementary Fig. 13a–c). Such a mechanism is reminiscent of that observed previously for the ErmDL arrest peptide, where the penultimate Arg sidechain of ErmDL NC also encroaches into the A-site binding pocket (Supplementary Fig. 13g–i)[32]. The major difference here is that ErmDL arrest requires the additional presence of a macrolide antibiotic bound within the NPET to evoke the arrest[32]. For CliM-SRC, we propose that, analogously to macrolides, it is the α-helical conformation and interactions of the N-terminal region of the CliM NC with the NPET that promote the non-productive conformation of the C-terminal region of CliM at the PTC. Indeed, our systematic in vivo mutational analyses demonstrated that introducing a Pro residue into this region generally compromised stalling (Fig. 3c), supporting both the presence and functional importance of this structural compaction in living cells. The conserved KYxIW motif is positioned between the constriction site and the PTC, and, therefore, interactions with the constriction site appear to mainly involve residues located further N-terminal to this motif. The importance of the interaction between CliM and uL22 is supported by both structural analysis and mutational studies (Fig. 6). Notably, the amino acid sequence of uL22 differs among *E. coli*, *B. subtilis*, and *C. difficile*, which may contribute to the species-specific arrest of CliM.

Considering that most arrest peptides stall exclusively at a single specific site, the remarkable flexibility in stalling site selection represents an unusual and defining feature of CliM. A prominent example of multi-site stalling is *B. subtilis* MifM, which contains an essential cluster of four acidic residues (DEED) near the PTC and undergoes arrest at four consecutive codons[22,33]. The consecutive appearance of acidic residues is proposed to enable interactions between the nascent chain and the ribosome to be maintained across multiple elongation cycles, thereby supporting multi-site stalling[33]. Only acidic residues in this region can support MifM-mediated stalling[33], which distinguishes it from CliM. The *Neurospora crassa* arginine attenuator peptide (AAP) provides an example of flexible stalling-site selection; relocating the stop codon toward the N-terminus results in premature termination arrest[34], although its mechanistic relevance to CliM remains unclear. The flexibility of CliM in stalling site selection, together with its ability to support both elongation and termination arrest, provides a mechanistic explanation for the seemingly distinct stalling behaviors of Ck and Cd CliM within a unified framework.

A noteworthy observation concerns the distinct mutational effects between the PTC-proximal and PTC-distal regions of CliM. Amino acid substitutions near the PTC exerted local, stalling site-specific effects, whereas mutations in the PTC-distal region had more global consequences. An example of the latter is the KWm, which abolished both −1 and 0-site stalling of Ck CliM (Fig. 2)[14]. Many mutations in PTC-distal residues of Cd CliM increased Spc resistance in the DMS analysis, whereas only a few were found in PTC-proximal residues (Fig. 3), further supporting this notion. The stronger conservation of amino acids in the PTC-distal region relative to the PTC-proximal region (Fig. 3c) may also reflect its global significance.

To date, three arrest peptides encoded upstream of *yidC* have been identified: *B. subtilis* MifM[8], actinomycetal ApcA[15], and the

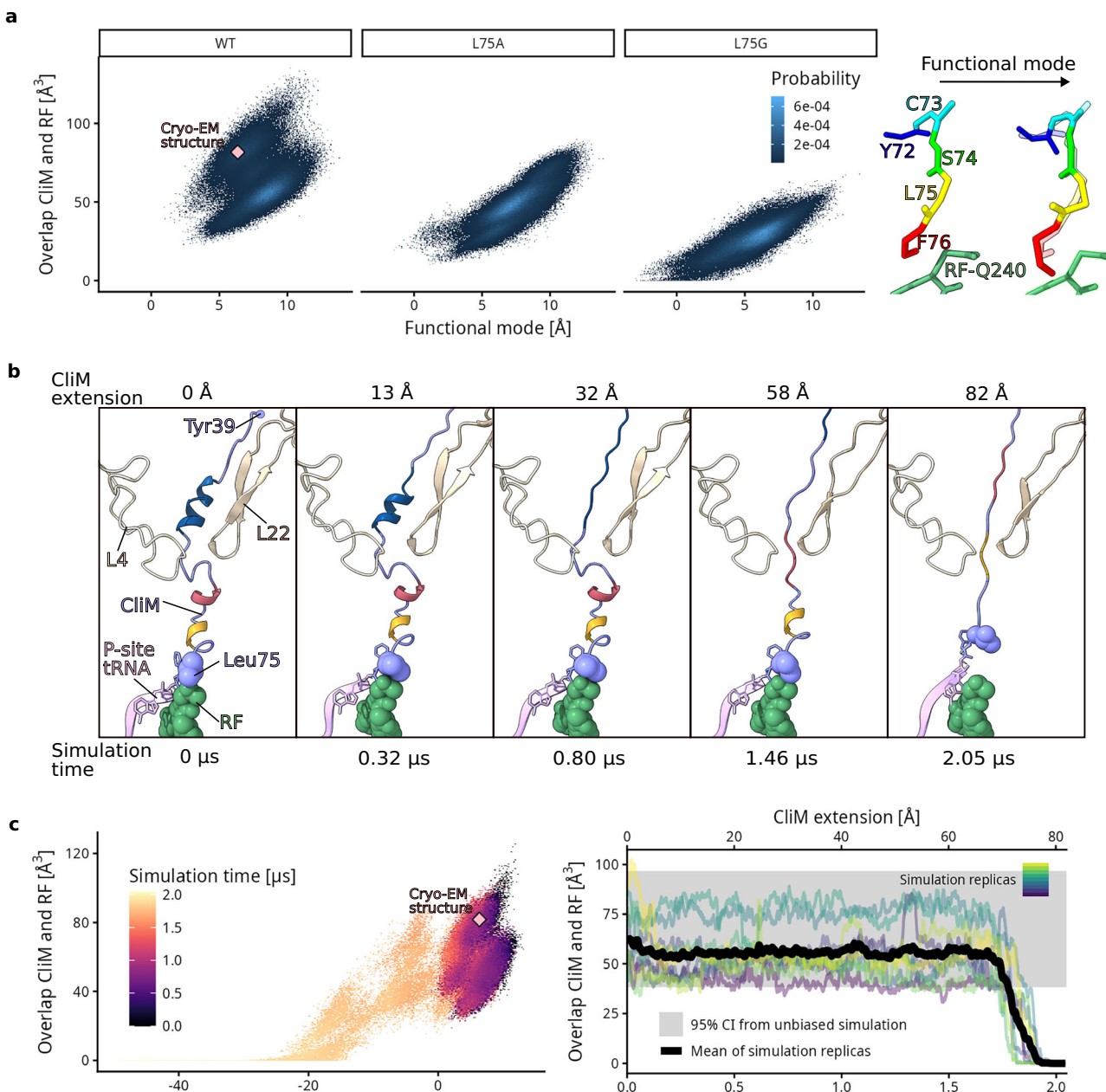

**Fig. 8 | MD simulations of stalling and its release. a** From equilibrium simulations of the CliM variants WT, L75A, and L75G, the functional mode of C-terminal backbone atoms that best predicts overlap volume between CliM and accommodated RF was identified. Left, histograms of overlap volume and the projection onto the functional mode are shown. The position of the CliM-SRC cryo-EM structure is indicated in pink. Right, the conformational change corresponding to the functional mode is shown. **b** Snapshots at different peptide extensions during one of the pulling simulations with pulling velocity 4 mm/s. **c** Left, projection onto the functional mode and overlap volume as a function of time during pulling. The color of each bin represents the average time stamp of the data points in that bin, taken from all pulling simulation replicas. Right, overlap volume during pulling for each replica, as well as the mean values. The gray rectangle indicates 95% CI from the WT equilibrium simulations.

clostridial CliM[14]. All of these originate from Gram-positive bacteria. This may be related to the higher frequency of *yidC* gene duplication observed in Gram-positive bacteria[8]. Among the three arrest peptides, only MifM had previously been demonstrated experimentally to monitor and regulate YidC[7,8]. In this study, we show that CliM also participates in sensing and regulating YidC levels. Genetic studies demonstrate that YidC-dependent membrane insertion relieves CliM-mediated translational arrest and that the mRNA secondary structure suppresses the expression of the downstream *yidC2* gene (Fig. 1). The spacing between the transmembrane segment and the arrest motif is conserved across homologs, both in MifM and CliM[14], supporting the

model that membrane insertion generates a pulling force that relieves arrest. Consistent with this notion, our MD simulations suggest that applying a pulling force onto the N-terminus of CliM triggers unwinding of the arrest-essential α-helical regions located in the PTC-distal region of the CliM NC (Fig. 9d), followed by dislodging of the sidechain of Leu75 from the A-site, thereby allowing accommodation of the GGQ-loop of the RF (Fig. 9d). The lack of sequence similarity between MifM and CliM makes it likely that independent evolution in each bacterial lineage has given rise to arrest peptides with distinct sequences and molecular mechanisms that nevertheless fulfill similar physiological roles.

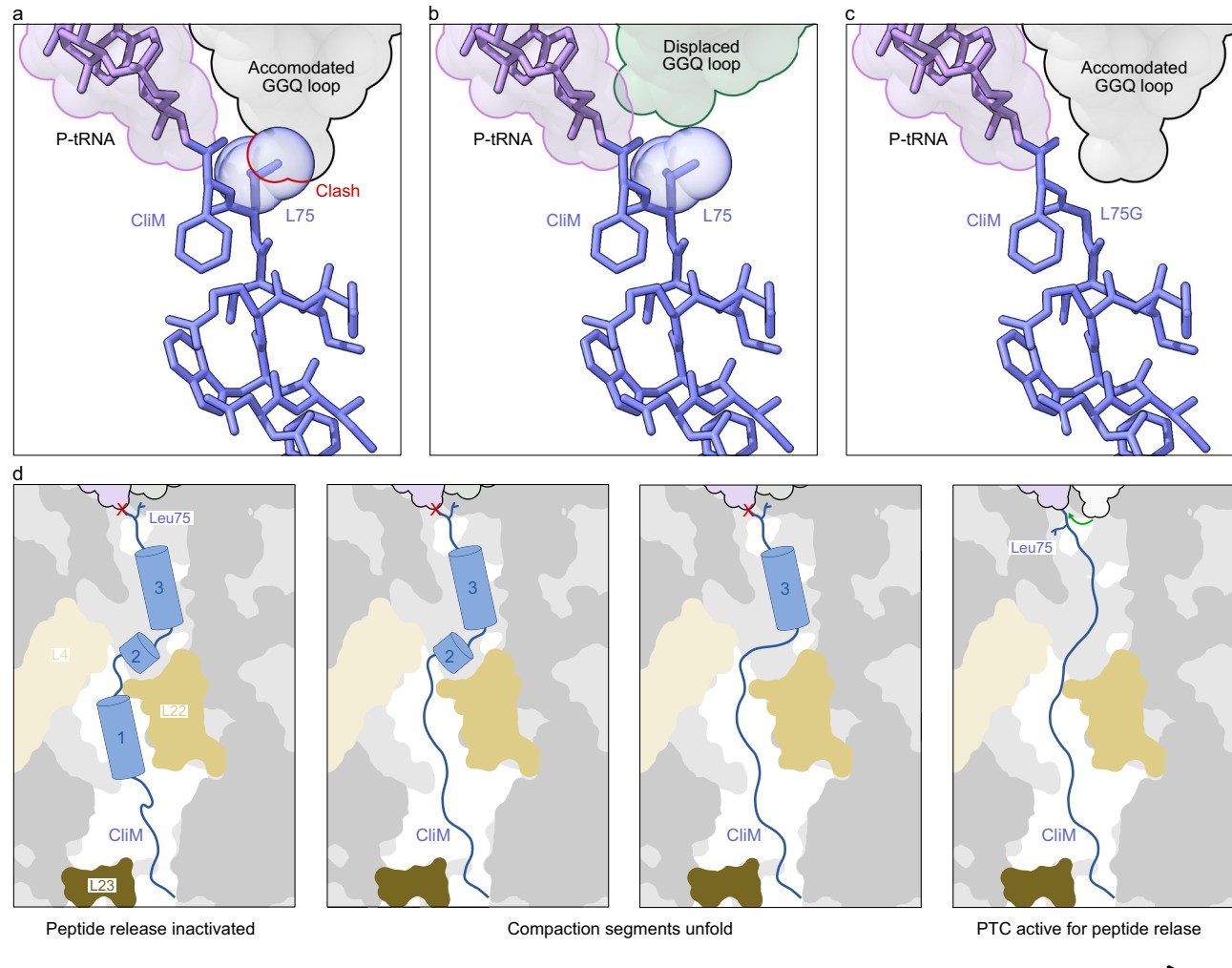

**Fig. 9 | Model for CliM-mediated translational stalling. a** An active, properly accommodated RF clashes with the position of the penultimate residue Leu75 of CliM, therefore, **b** the GGQ loop of the RF in CliM is displaced, whereas **c** a L75G version of CliM showed to lose its stalling capacity since it allows the RF to properly accommodate in the A-site of the PTC. **d** When a pulling force is applied to the nascent chain, individually secondary structured domains unfold piece by piece from the exit of the tunnel to the PTC until the penultimate Leu75 of CliM is displaced, thereby allowing proper accommodation of the RF in the A-site of the PTC and release is facilitated.

Labels within figure d: Peptide release inactivated | Compaction segments unfold | PTC active for peptide relase

Change in nascent chain compaction over time by application of pulling force

## Methods

### Bacterial strains and plasmids

The *B. subtilis* strains, plasmids, and DNA oligonucleotides used in this study are listed in Supplementary Tables 1–3, respectively. Plasmids were constructed via standard cloning methods, including PCR using PrimeSTAR GXL (Takara) and Gibson assembly[35]. Sera-Mag Carboxylate-Modified Magnetic Particles (Cytiva, 65152105050250) were used to purify double-stranded DNA and sequencing products. The *B. subtilis* strains were constructed by transformation involving double homologous recombination between chromosomal DNA and the plasmids introduced into *B. subtilis* competent cells. The resulting recombinant clones were validated based on their antibiotic-resistance markers.

### Culture media and growth conditions

*B. subtilis* cells were cultured in LB medium. *E. coli* cells were cultured in LB medium supplemented with 100 µg/ml ampicillin. Cells were cultured at 37 °C and collected for Western blotting or β-galactosidase activity assay when they reached an optical density of 0.5–1.0 at 600 nm (OD$_{600}$).

### Purification of *B. subtilis* and *C. difficile* ribosomes

*B. subtilis* strain PY79 was grown at 37 °C in LB medium until mid-log phase. Cells were then harvested by centrifugation, washed twice with buffer I-H (10 mM HEPES-KOH pH 7.6, 15 mM Mg(OAc)$_2$, 1 M KCl, 5 mM EDTA, 10 mM 2-mercaptoethanol), then once with buffer II-H (Buffer I-H with 50 mM KCl) and stored as a pellet at −80 °C. They were thawed in suspension buffer (10 mM HEPES-KOH pH 7.6, 50 mM KCl, 10 mM Mg(OAc)$_2$, 7 mM 2-mercaptoethanol) with protease inhibitors (1 mg/ml PefaBlock, 10 µg/ml Leupeptin, 0.7 µg/ml Pepstatin) and disrupted by passing through a microfluidizer LV1 (Microfluidics) at 16,000 psi. Cell lysate was mixed with an equal volume of suspension buffer with 3 M ammonium sulfate and kept on ice for 30 min. Cell debris and protein precipitates were removed by centrifugation (4 °C, 10,000 × *g*, for 30 min). The supernatant was subjected to HiTrap Butyl FF column of 10 ml volume (GE) equilibrated with buffer A (20 mM HEPES-KOH pH 7.6, 1.5 M (NH$_4$)$_2$SO$_4$, 10 mM Mg(OAc)$_2$, 7 mM β-mercaptoethanol). The column was then washed with 40 ml of buffer B (20 mM HEPES-KOH pH 7.6, 10 mM Mg(OAc)$_2$, 7 mM β-mercaptoethanol) containing 1.2 M ammonium sulfate and proteins were eluted with a 1.2 M to 0 M gradient of ammonium sulfate in buffer B. Fractions containing the ribosome were combined and loaded on the top of an equal volume of 30%

sucrose cushion buffer (20 mM HEPES-KOH pH 7.6, 10 mM Mg(OAc)$_2$, 30 mM NH$_4$Cl, 30% sucrose, 7 mM 2-mercaptoethanol) and ultra-centrifuged (4 °C, 100,000 × $g$ for 16 h) to sediment the ribosomes. The pellet was suspended with ribosome buffer (20 mM HEPES-KOH pH 7.6, 30 mM KCl, 6 mM Mg(OAc)$_2$, 7 mM β-mercaptoethanol) and stored at −80 °C until use for in vitro translation assays. *C. difficile* ribosome was also purified according to the above procedure with some modifications. *C.difficile* strain 630 Δ*erm* was grown anaerobically in Brain-Heart Infusion (Difco) supplemented with 0.5% yeast extract and 0.1% cysteine (BHIS) under 80% N$_2$, 20% CO$_2$, 4% H$_2$ atmosphere in the Coyanaerobic chamber (COY). Cells were then harvested by centrifugation and stored as a pellet at −80 °C. They were thawed in suspension buffer (10 mM HEPES-KOH pH 7.6, 50 mM KCl, 10 mM Mg(OAc)$_2$,7 mM 2-mercaptoethanol). The cell suspension was passed twice at 150,000 psi through a French Press 40 K cell (Amicon, USA) and centrifuged (20 min, 8000 × $g$, 4 °C) to remove cell debris, resulting in the crude extract. The crude extract was loaded onto sucrose cushions (1.1 M sucrose, 500 mM NH$_4$Cl, 15 mM Mg(OAc)$_2$, 0.5 mM EDTA, 3 mM β-mercaptoethanol, 20 mM Tris:Hcl pH 7.5) and ultracentrifuged (4 °C, 100,000 × $g$, for 18 h). The pellets were dissolved in HEPES: Polymix buffer (20 mM HEPES: KOH pH 7.5, 2 mM DTT, 15 mM Mg(OAc)$_2$, 95 mM KCl, 5 mM NH$_4$Cl, 0.5 mM CaCl$_2$, 8 mM putrescine, 1 mM spermidine) and snap-frozen in liquid nitrogen and stored at −80 °C. To remove residual proteins, the thawed sample was mixed 1:1 with suspension buffer containing 3 M ammonium sulfate and incubated on ice for 30 min. Precipitates were removed by centrifugation (10,000 × $g$, 30 min, 4 °C). The supernatant was applied to a HiTrap Butyl FF column (GE) following the same procedure as used for *B. subtilis* ribosome purification.

### In vitro translation and Western blotting

Bacterial reconstituted transcription–translation coupling systems[19,20] were used in the in vitro translation assay. For in vitro translation using *E. coli* ribosomes, we utilized PUREfrex version 1.0 (GeneFrontier). For Bs PURE, purified *B. subtilis* ribosomes were used at the final concentration of 1 µM in the PURE system, without adding *E. coli* ribosomes. 2.5 U/µL of T7 RNA polymerase (Takara) was added to ensure transcription. The in vitro translation reaction was primed using the DNA templates prepared by consecutive PCR as follows. The first PCR products were amplified from respective plasmids using primers PT7-RBSkf-GFP (TAACTTTAAGAAGGAGGGAGATATACCAATGACAATGTTT GTGGGATC) and lacZ60-TAATAA-21-rv (TGGTGCCGGAAACCAGG-CAAATTATTAGCGCCATTCGCCATTCAGGCT), and then used as a template for the second PCR with primers Universal primer-77-PURE (GAAATTAATACGACTCACTATAGGGAGACCACAACGGTTTCCCTCTA-GAAATAATTTTGTTTAACTTTAAGAAGGAG) and lacZ60-TAATAA-21-rv. The translation reaction was carried out for 30 min at 37 °C and was stopped by adding 2 × SDS–PAGE loading buffer, for Western blotting. A portion of the sample was further treated with 0.2 mg/ml RNase A (Promega) at 37 °C for 10 min, to degrade the tRNA moiety of the peptidyl-tRNA. The translation products were separated on a 10% polyacrylamide gel that was prepared using WIDE RANGE Gel buffer (Nacalai Tesque), according to the manufacturer's instructions, then transferred onto a PVDF membrane (Merck, IPVH00010) and subjected to immuno-detection using antibodies against GFP (mFX75; Wako, 012-22541). Images were acquired and analyzed using an Amersham Imager 600 luminoimager (GE Healthcare), and the band intensity was quantified using ImageQuant TL (GE Healthcare).

### Toeprinting assay

In vitro translation was carried out using the Bs PURE or Cd PURE system at 37 °C for 20 min in the presence or absence of 0.1 mg/mL chloramphenicol, a translation inhibitor. The translation reaction mixture was then mixed with the same volume of the reverse transcription mixture containing 50 mM HEPES-KOH, pH 7.6, 100 mM

potassium glutamate, 2 mM spermidine, 13 mM magnesium acetate, 1 mM DTT, 2 µM of oligonucleotide labeled with 6-carboxyfluorescein (6-FAM) at the 5′ end (5′−AGGGCGATCGGTGCGGGCCTCTTC−3′, Invitrogen), 50 µM each dNTP, and 10 U/µL ReverTra Ace (Toyobo), then incubated further at 37 °C for 15 min. The reaction mixture was diluted 5-fold with the NTC buffer (Macherey-Nagel), and the reverse transcription products were purified using a NucleoSpin Gel and PCR Clean-up kit (Macherey-Nagel). The reverse transcription products were eluted with 30 µL of HiDi formamide (Thermo Fisher). Samples were then mixed with 10 µL of 10-fold-diluted GeneScan 500 LIZ dye size standard (Thermo Fisher, 4322682), then heated at 96 °C for 3 min just before capillary electrophoresis. The dideoxy DNA samples used as size markers for sequencing were prepared using a Thermo Sequenase Dye Primer Manual Cycle Sequencing Kit (Thermo, 79260), Thermo Sequenase Cycle Sequencing Kit (Thermo, 785001KT), or Thermo Sequenase DNA Polymerase (Cytiva, E79000Y), according to the manufacturer's instructions, with some modifications. The DNA polymerase reaction was carried out using the same sets of template DNA and primer used for the toeprint assay. Each reaction mixture contained 0.44 µM of the 6-FAM-labeled primer, 60 µM each deoxynucleotide triphosphate (dATP, dCTP, dGTP, and dUTP), and 0.6 µM dideoxynucleotide triphosphate (either ddATP, ddCTP, ddGTP, or ddUTP). The sequencing products were purified using Sera-Mag Speed Beads and eluted with HiDi formamide. Next, 2 µL of a 10-fold-diluted GeneScan 500 LIZ dye size standard was added. If needed, the toeprinting product was further diluted before electrophoresis using HiDi formamide. The toeprinting and dideoxy sequencing products were then subjected to fragment analysis on a SeqStudio genetic analyzer (Thermo Fisher). Fragment data were analyzed using the GeneMapper software version 6 (Applied Biosystems) and shown as a gel-style heatmap based on the fragment size and signal intensity.

### In vivo β-galactosidase assay

A 100-µL aliquot of the culture was transferred to a well in a 96-well plate, and OD$_{600}$ was recorded. We mixed the culture with 50 µL of Y-PER reagent (Thermo Fisher) for 20 min at room temperature, to disrupt the cells. Subsequently, 30 µL of *o*-nitrophenyl-β-D-galacto-pyranoside (ONPG) in Z-buffer (60 mM Na$_2$HPO$_4$, 40 mM NaH$_2$PO$_4$, 10 mM KCl, 1 mM MgSO$_4$, and 38 mM 2-mercaptoethanol) was added to the cell lysate, and the OD$_{420}$ and OD$_{550}$ were measured at 28 °C every 5 min over a period of 60 min. Arbitrary units of β-galactosidase activity were calculated using the following formula: $[(1000 × V_{420} − 1.3 × V_{550})/OD_{600}]$, where $V_{420}$ and $V_{550}$ are the first-order rate constants, OD$_{420}$/min and OD$_{550}$/min, respectively.

### Deep mutational scanning of Cd CliM

A SpcR-reporter plasmid (*gfp-Cd_cliM-spcR*: pCH2674) was constructed by fusing a gene fragment encoding residues 30–76 of Cd CliM in-frame with *gfp* and *flag-spcR*, with the native stop codon of *cliM* replaced by a Lys codon. Using degenerate primers (Supplementary Table 4), we introduced comprehensive single-residue substitutions at positions 38–76 as well as at the artificially introduced Lys77 codon. The resulting plasmid library was integrated into the chromosome of *B. subtilis*, pooled, and stored as glycerol stocks at −80 °C.

Aliquots of the library were inoculated into LB medium without spectinomycin (Spc) and cultured to mid-log phase (OD600 = 0.5, defined as t0). Cells were then subcultured into LB medium with or without Spc (100 or 250 µg/mL) and grown again to mid-log phase (t1). At both t0 and t1, cells were collected for genomic DNA extraction and a portion of the cultures was serially diluted and plated on LB agar plates to determine colony-forming units (CFU). Genomic DNA was extracted as follows. Cells were resuspended in 200 µL buffer G3 (NucleoBond Buffer Set III, Takara, Cat. No. 740603), supplemented with 4 µL of lysozyme solution (100 mg/mL; Sigma Cat. No. L6876) and 5 µL of NucleoBond Proteinase K (provided in the kit), and incubated at

37 °C for 20 min. Subsequently, 80 µL buffer G4 was added, vortexed, and incubated at 50 °C for 30 min. To this lysate, 200 µL SeraPURE beads were added to capture DNA. After two washes with 70% ethanol, DNA was eluted with 200 µL MilliQ water.

For NGS library preparation, DNA was PCR-amplified using primers listed in Supplementary Table 5. The PCR mixture contained 2 µL chromosomal DNA (5 ng/µL) in 40 µL GXL Fast reaction mixture (Takara). PCR was carried out for 24 cycles under the following conditions: 98 °C for 10 s, 55 °C for 5 s, and 68 °C for 20 s. PCR products were purified with SeraPURE beads and eluted in 10 µL EB buffer. Purified products were separated on a 10% polyacrylamide gel (Fujifilm, SuperSep Ace 13 well) using 1 × TBE buffer, and the PCR bands were excised from the gel. The gel slices were homogenized with a pestle in 300 µL DNA elution buffer (10 mM Tris-HCl, pH 8.0, 300 mM NaCl, 1 mM EDTA) and vortexed overnight at 37 °C at 15,000 rpm to extract DNA. The homogenates were applied to an EconoSpin column (Gene Design) and centrifuged at 11,000 rpm for 3 min at 4 °C to remove gel debris, while recovering DNA in the flow-through. The flow-through was mixed with 600 µL NTC buffer and applied to a NucleoSpin Gel and PCR Cleanup column (Macherey-Nagel). After two washes with NT3 buffer, DNA was eluted with 17 µL NE buffer. NGS was outsourced to Rhelixa (Tokyo, Japan) and performed using the Illumina NovaSeq X Plus platform.

## DMS data analysis

The DNA libraries for Illumina sequencing were constructed with the following structure: 5′-*AATGATACGGCGACCACCGAGATCTACACAGCGCTAGACACTCTTTCCCTACACGACGCTCTTCCGATCT*NNNNNN**TAG**AGAGACCACATGGTCCTTCTTGAGTTTGTAACAGCTGCTGGGATTACACATGGCATGGATGAACTATACAAAAAAGACCTCTTAAATCATAAAATTAA<u>GTATGTTTTAATAAGAGACATATTTGTAAATAGAATTACATATTCTGAAGAACGACTGCCTAAACAGTATATAGTTTTTCAGAAATATGATATTTGGCGGTATTGTAGTTTATTT</u>AAAGACTATAAAGACGACGACGACAA**A**GGT**NNNNNN**AGATCGGAAGAGCACACGTCTGAACTCCAGTCACCCGCGGTTATCTCGTATGCCGTCTTCTGCTTG-3′. Both terminal regions indicated in italics correspond to Illumina adapters: the 5′ end contains the P5 adapter, the i5 index, and the binding site for Read 1 sequencing primer, while the 3′ end contains the P7 adapter, the i7 index, and the binding site for Read 2 sequencing primer. Each "NNNNNN" represents a six-nucleotide randomized sequence. Within these regions, the trinucleotide TAG (designated bc5) and the trinucleotide GGT (designated bc7) indicated in bold serve as barcodes. The intervening sequence between bc5 and bc7 constitutes the reporter sequence, and the underlined sequence corresponds to codons 38–76 of wild-type *cliM(Cd)* and the artificially introduced Lys77 codon, in which each codon was replaced with an NNK random codon. Sequencing was carried out with 150-bp paired-end reads. The forward and reverse reads overlapped by 42 bp, allowing precise determination of the *cliM* region. Raw sequencing reads were demultiplexed according to their barcode sequences using Seqkit (version 2.10.0)[36]. Subsequently, paired-end reads were merged into single reads based on their overlapping regions using fastp (version 0.24.0)[37]. Quality filtering was also performed with fastp using the following options: −q 20 (minimum Phred quality score for a qualified base), −u 30 (maximum percentage of unqualified bases per read), and −e 25 (minimum average quality score for a read). Reads were then assigned to each mutated sequence variant and counted using R (version 4.5.0) with the packages ShortRead (version 1.66.0)[38] and tidyverse (version 2.0.0)[39]. The growth rate of each variant was calculated using the following equation:

$$\text{growth rate} = \log_2[(\text{CFU}_{t1}/\text{CFU}_{t0}) \times (\text{RPM}_{t1}/\text{RPM}_{t0})]/\text{culture hour},$$

where *CFU* represents colony-forming units, *RPM* represents reads per million, and *Culture hour* indicates the incubation time between *t0* and *t1*. The relative fitness under 250 µg/mL spectinomycin (Spec) was calculated by dividing the growth rate under the antibiotic condition by that under the no-antibiotic condition. The experiments were performed independently twice, and the mean relative fitness values were plotted as a heatmap using only variants with at least eight reads. The reproducibility between biological replicates is shown in Supplementary Fig. 2a, b and Source Data 2.

## Search for CliM homologs

Representative genomes belonging to the class Clostridia were identified from the GTDB representative genome set (release 226)[40] and their genome sequences were downloaded from NCBI using the datasets command-line tool (version 16.34.1)[41]. A total of 15,460 downloaded genomes were translated in all six reading frames using seqkit (version 2.10.0)[36]. From the translated sequences, proteins homologous to Cd CliM (protein ID: WP_009891651.1) or Cd YidC2 (WP_003434300.1) were retrieved using HMMER jackhmmer (version 3.4)[42] with a maximum of five iterations and an *E*-value threshold of 1e−3. Among the CliM-like candidates, only those encoded upstream of *yidC* were retained and designated as CliM homologs (listed in Source Data 1). The CliM homolog sequences were aligned using the MAFFT einsi algorithm (version 7.511)[43]. Regions comprising 28 amino acid residues upstream and 11 residues downstream of the Lys position within the KWxIW motif were extracted and used to generate sequence logos with ggseqlogo (version 0.2)[44].

## Plasmid

Cell extracts for in vitro translation on *B. subtilis* ribosomes were generated as described previously[22] but using the *B. subtilis* strain 168 Δ*yuyD* Δ*ssrA* Δ*smpB* (see below). The protein coding sequence of CliM from *C. difficile* strain 630 was cloned into pDG1662 downstream of a T7 promoter, a ribosome binding site, a His-tag and a FLAG-tag using restriction enzyme SphI and HindIII (NEB) and T4 ligase (NEB). The insert of CliM was amplified by PCR using Q5 High-Fidelity DNA polymerase (NEB) from the *E. coli* strain K12 genome using as primers Fwd_SphI_CliM (5′-TTTTTTGCATGCAGATTAGATATGGATAGTATGTTTATATGTAC-3′) and Rev_CliM_stop_HindIII (5′-AAAAAAAAGCTTTTAAAACAAACTACAATACCTCC-3′). DNA oligo primers used were purchased from Metabion. Cloning was performed as recommended by Sambrook et al.[45].

## PCR and in vitro transcription

PCR reaction (Q5 High-Fidelity DNA Polymerase (NEB), Q5 Reaction buffer (NEB)) was used with primers M13 fwd (5′-GTAAAACGACGGCCAGT-3′) and M13 rev (5′-CAGGAAACAGCTATGAC-3′) on the vector harboring CliM ORF to generate the amplified DNA sequence (5′-CAGGAAACAGCTATGACCATGATTACGGAATTCGAGCTCGGTACCCGGGATCCCGCGAAATTAATACGACTCACTATAGGGGAATTGTGAGCGGATAACAATTCCCCACTAGTAATAATTTTGTTTAACTTTAAGAAGGAGATATACCATGGGCAGCAGCCATCATCATCATCATCACGATTACAAGGATGACGACGATAAGGCTAGCAGCAGCGGTACCGGCAGCGGCGAAAACCTCTATTTTCAGGGTAGTGCGCAAGCATGCAGATTAGATATGGATAGTATGTTATATGTACTAATTTATATGACATTATTGGTATCTATCATTGGAAGTATATTTTATTTTTGTAAAGACCTCTTAAATCATAAATTAAGTATGTTTTAATAAGAGACATATTTGTAAATAGAATTACATATTCTGAAGAACGACTGCCTAAACAGTATATAGTTTTTCAGAAATATGATATTTGGAGGTATTGTAGTTTGTTTTAAAAGCTTGGACTGGCCGTCGTTTTAC-3′; underlined are the T7 promoter region, ribosomal binding site, start codon, FLAG-tag and stop codon, respectively). PCR conditions applied were as suggested by the manufacturer and PCR products were purified via spin columns, and in vitro transcription reaction was set up using 1 µg PCR product per 50 µl reaction volume and T7 RNA polymerase (Thermo Scientific™). RNA was purified by LiCl precipitation and washed with ethanol.

### Bacillus subtilis S12 translation extract

The *B. subtilis* S12 translation extract was prepared following a procedure described[22], with some modifications. Briefly, cells (*B. subtilis* strain 168 Δ*yvyD* Δ*ssrA* Δ*smpB*) were grown to OD$_{600}$ 0.8 in 2× YPTG medium (16 g L$^{-1}$ peptone, 10 g L$^{-1}$ yeast extract, 5 g L$^{-1}$ NaCl, 22 mM NaH$_2$PO$_4$, 40 mM Na$_2$HPO$_4$, 19.8 g L$^{-1}$ glucose, sterile-filtered) at 37 °C. Cells were collected by centrifugation at 5000 rpm at room temperature for 15 min and subsequently washed 3× in room temperature Buffer A (10 mM Tris–acetate (pH 8.2, 4 °C), 14 mM magnesium acetate, 60 mM potassium glutamate, 1 mM dithiothreitol, and 6 mM 2-mercaptoethanol, sterile-filtered). After the third wash, cells were resuspended in a minimal volume (0.7 mL g$^{-1}$) of room temperature Buffer B (Buffer A without 2-mercaptoethanol). Cells were snap-frozen in liquid nitrogen and stored at −80 °C. Cells were subsequently thawed on ice and then lysed using FastPrep−24™ MP (4 × 30 min, shaking 4.5 m s$^{-1}$ intercalated by 1 min rest on ice), the lysate was collected by centrifugation (1000 × *g*, 4 °C, 1 min) and further cleared by centrifugation at 12,000 × *g*, 4 °C, 10 min. The lysate was used immediately, or aliquoted, snap-frozen, and stored at −80 °C.

### Generation of the complexes

To generate the CliM-SRC, the CliM mRNA template (500 ng µL$^{-1}$) was translated by incubation in a *B. subtilis* in vitro translation system. Briefly, a total reaction volume of 800 µL was prepared by mixing 208 µL reconstitution buffer, 16 µL of methionine, 160 µL amino acid mix, 136 µL reaction mix (from RTS 100 HY Kit from Biotechrabbit GmbH) with 160 µL *B. subtilis* S12 translation extract, 40 µL CliM mRNA, 32 µL 300 mM magnesium acetate, 48 µL DEPC-treated H$_2$O, and then incubated for 30 min at 30 °C shaking in a thermomixer (500 rpm).

### Purification of the complexes for structural analysis

The complex previously generated was purified by incubating the reaction with 100 µL anti-FLAG® M2 affinity gel (Merck), previously equilibrated with Hico buffer (50 mM HEPES-KOH (pH = 7.4, 4 °C), 100 mM potassium acetate, 15 mM magnesium acetate, 1 mM dithiothreitol, 0.01 % (w/v) n-dodecyl-beta-maltoside, sterile-filtered) inside Mobicol column provided with 35 µm filter (MoBiTec) at 4 °C for 3.5 h while rolling. After removal of the flowthrough, the beads were washed with a total of 2 mL Hico buffer and then the bound complex was eventually eluted by incubation with 30 µL Hico buffer containing 0.6 mg/mL 3XFLAG peptide for 45 min at 4 °C while rolling, followed by centrifugation (2000 × *g*, 4 °C, 2 min). Aliquots from each fraction were checked by western blotting.

### Cryo-EM sample preparation

3.5 µl of sample (8.4 OD260/ml) was applied to grids (Quantifoil, Cu, 300 mesh, R3/3 with 3 nm carbon, Product: C3-C18nCu30-01) which had been freshly glow discharged using a GloQube® Plus (Quorum Technologies) in negative charge at 25 mA for 30 s to make grids hydrophilic. Sample vitrification was performed using a mixture of ethane/propane in a 1:2 ratio in a Vitrobot Mark IV (Thermo Scientific), the chamber was set to 4 °C and 100% rel. humidity, blotting was done for 3 s with zero blot force with Whatman 597 blotting paper. The grids were subsequently clipped in autogrids cartridges and stored in liquid nitrogen until high-resolution data collection.

### Cryo-EM data collection

Data collection was performed on 300 kV Titan Krios (Thermo Fisher/FEI) with Fringe-Free Imaging (FFI) setup and equipped with Gatan K3 direct electron detector using EPU (version 3.2.0.4775REL). Magnification of ×105,000 was used, with data collected using super resolution counted mode at 0.415 pixel size, binned twice on the fly through EPU, yielding 0.83 pixel size. Total 40 e/A$^2$ fluence was fractionated into 35 frames, resulting in 1.14 e/A$^2$ dose per frame and total exposure

of 1.91 s in Nanoprobe mode (15 e$^-$/px/s over an empty area on the camera level). Defocus range of −0.3 to −0.9 was used with a step size of 0.1 between holes. C2 aperture of 70 µm was inserted with a beam spot size of 7. A BioQuantum energy filter set to a 20 eV cut-off was used to remove inelastically scattered electrons. Final objective astigmatism correction <1 nm and auto coma free alignment <40 nm was achieved using the AutoCTF function of Sherpa (version 2.11.1). A total of 11,409 micrographs were collected for CliM-SRC (12 exposures per hole) and saved as tiff gain-corrected files.

### Single-particle reconstruction of SRC complexes

RELION (version 4.0)[46,47] was used for processing, unless otherwise specified. For motion correction, RELION's implementation of MotionCor2 with 4 × 4 patches, and, for initial contrast transfer function (CTF) estimation, CTFFIND version 4.1.14[48], were employed.

From 11,409 micrographs, 1,204,971 particles were picked using RELION's implementation of Auto-picking with a particle diameter of 200–300 Å. In total, 854,071 ribosome-like particles were selected after two-dimensional (2D) classification and extracted at 2× decimated pixel size (1.662 Å per pixel) (Supplementary Fig. 3). An initial three-dimensional (3D) consensus refinement was done using a molmap of *B. subtilis* 70S ribosome model (PDB ID 6HA8)[49] without tRNAs, followed by partial signal subtraction on the particles with a mask around tRNAs sites to perform focussed classification with 5 classes. Two classes contained 70S ribosomes with P-tRNA (224,958 and 74,875 particles), two classes contained 70S ribosomes with P-tRNA and release factor (228,372 and 241,319 particles) and one class contained 70S ribosomes with P-tRNA and A-tRNA (84,547 particles) (Supplementary Fig. 3c). The latter was subsorted with a mask around the A-site into four classes which yielded one class containing P-tRNA and release factor (36,541 particles) and one class containing P-tRNA and A-tRNA (47,146 particles) (Supplementary Fig. 3d). The two classes containing P-tRNA and release factor were also further classified into two classes with a mask around the A-site, yielding one class with P-tRNA and release factor (234,795 particles) and one class with P-tRNA and substoichiometric release factor (Supplementary Fig. 3d). Both classes were again subclassified with a mask around the A-site leading to three classes containing P-tRNA and release factor (89,151, 145,644, and 76,887 particles) and one class with only P-tRNA (158,009 particles). All classes containing P-tRNA only were joined and subclassified with a mask around the A-site into 3 classes, leading to two classes of high-resolution containing P-tRNA (180,984 and 187,449 particles) (Supplementary Fig. 3e). Particles of the three major classes (P-tRNA and A-tRNA, P-tRNA only and P-tRNA and release factor) were individually joined for further processing. In particular, the resulting classes´ subtracted particles were reverted to their original images and 3D refined and CTF refined (4th order aberrations, beam tilt, anisotropic magnification and per-particle defocus value estimation), then subjected to Bayesian polishing[50] and another round of CTF refinement. For the CliM-SRC with P-tRNA and A-tRNA, a final resolution (gold-standard FSC0.143) of masked reconstructions of 2.8 Å was achieved (Supplementary Fig. 3f); for the CliM-SRC with P-tRNA only, a final resolution (gold-standard FSC0.143) of masked reconstructions of 2.3 Å was achieved (Supplementary Fig. 3g); and for the CliM-SRC with P-tRNA and release factor, a final resolution (gold-standard FSC0.143) of masked reconstructions of 2.3 Å was achieved (Supplementary Fig. 3h and Supplementary Fig. 4).

To estimate local resolution values, Bsoft[51] was used on the half-maps of the final reconstructions (blocres -sampling 0.831 -maxres -box 20 -cutoff 0.143 -verbose 1 -fill 150 -origin 0,0,0 -Mask half_map1 half_map2) (Supplementary Fig. 4d–f, h, j).

### Molecular modeling of SRC complexes

The molecular models of the 30S and 50S ribosomal subunits were based on the *B. subtilis* 70S ribosome (PDB ID 6HA8)[49] for CliM-SRC.

The tRNAs and nascent chains were modeled de novo. Restraint files for modified residues were created using aceDRG[52]. Starting models were rigid body fitted using ChimeraX[53] and modeled using Coot 0.9.8.92[54] from the CCP4 software suite version 8.0[54]. The sequence for the tRNAs was adjusted based on the appropriate anticodons corresponding to the mRNA. Final refinements were done in REFMAC 5 (v5.8.0415)[55] using Servalcat (v0.3.1)[56]. The molecular models were validated using Phenix comprehensive cryo-EM validation in Phenix 1.20–4487[57].

## MD simulation setup

For the MD simulations of WT CliM, a starting structure was obtained by extracting all residues within 35 Å of the CliM nascent chain from the structure of CliM-SRC with vacant A-site. To obtain starting structures for the L75A and L75G variants, we mutated Leu75 to alanine and glycine in PyMOL (The PyMOL Molecular Graphics System, Version 3.0, Schrödinger, LLC). The protonation states of histidine residues were determined with the WHATIF software[58]. Structural ions were added from an aligned high-resolution cryo-EM structure (PDB id: 8QOA)[23]. The resulting structures were then placed in a triclinic box with a minimum distance to the box boundaries of 15 Å. The principal components of the CliM nascent chain were oriented with the coordinate axes, so that the longest CliM axis was aligned with the x-axis. To accommodate an extended nascent chain in the pulling simulations, the box was extended along the x-axis, resulting in dimensions (x,y,z) of 211.13, 132.56, and 135.18 Å. Next, the system was solvated with water molecules using the program SOLVATE[59]. Subsequently, 7 mM MgCl and 150 mM KCl were added using GENION[59] before neutralizing the system with K+ ions.

All simulations were conducted with GROMACS 2025[59], together with the Amber ff14sb force field[60]. We used the $K^+$ and $Cl^-$ parameters from Joung and Cheatham[61] and the microMg parameters from Grotz et al. for Mg+ ions[62]. Lennard-Jones and short-range electrostatic interactions were computed up to a cut-off distance of 0.8 nm. Particle-mesh Ewald summation with a grid spacing of 0.12 nm was used to calculate long-range electrostatic interactions at larger distances[63]. We constrained all bond lengths with the LINCS algorithm[64] and used virtual sites for hydrogen atoms[65], allowing for an integration time step of 4 fs. Solvent and solute were independently coupled to a heat bath at 300 K using stochastic velocity rescaling[66] with a coupling time constant of 0.1 ps. The pressure was coupled to a stochastic cell rescaling barostat[67] at 1 bar with a time constant of 5 ps. Solute atom coordinates were recorded every 5 ps.

After performing an energy minimization with position restraints applied to all heavy solute atoms ($k = 1000$ kJ mol$^{-1}$ nm$^{-2}$), we equilibrated the solvent in the NPT thermodynamic ensemble for 50 ns using the same position restraints. This step was followed by a 20 ns simulation, where the position restraints on atoms within 25 Å around the CliM nascent chain were linearly decreased to zero. The position restraints on atoms in the outer layer (25–35 Å) were linearly changed to force constants calculated from the atomic root mean square fluctuations from a previous whole-ribosome simulation[68].

Then we recorded 2 μs long non-biased production trajectories for WT, L75A and L75G CliM each. The equilibration and production runs were repeated for a total of 10 replicas per system, each starting from a different initial Boltzmann distribution of atom velocities sampled at 300 K.

In addition to the equilibrium simulations, we also simulated pulling on the nascent chain for the WT CliM. To this end, we added a harmonic potential, centered at the center of mass of the N-terminal residue of the nascent chain, with a force constant of $k = 1000$ kJ mol$^{-1}$ nm$^{-2}$. Over the course of the simulation, this potential was moved along the x-axis (direction of the exit tunnel) at a constant velocity for a total distance of 82 Å. We simulated pulling at different velocities, proportional to the total simulation time of 64, 128, 512, 1024, and 2048 ns.

## MD simulation analysis

Prior to analysis, all frames of the MD trajectories were aligned to the cryo-EM structure: First, a structural alignment of the RNA phosphate backbone in the simulation input structures and cryo-EM model was performed with PyMOL. Then, a rigid-body fit between the phosphate backbone of each trajectory frame and the aligned input structure was carried out.

The overlap volume between CliM and the release factor was calculated by using a Monte–Carlo volume estimation algorithm as described previously in Beckert et al.[32].

For functional mode analysis, non-biased trajectories of WT, L75A and L75G CliM were concatenated, and the backbone atoms of the 5 C-terminal CliM residues were selected for analysis[69]. Half of the replicas of each variant were used to train the model, and the other half were used for validation.

## Figures

UCSF ChimeraX (v1.9) was used to isolate density and visualize density images and structural superpositions. Models were aligned using the Matchmaker function in ChimeraX 1.9. Figures with MD simulation results were created with R 4.5.1 and ChimeraX 1.10.1. Figures were assembled with Inkscape 1.3.

## Reporting summary

Further information on research design is available in the Nature Portfolio Reporting Summary linked to this article.

## Data availability

The raw sequencing reads from the deep mutational scanning have been deposited in the DDBJ under BioProject accession number PRJDB39765. Micrographs have been deposited as uncorrected frames in the Electron Microscopy Public Image Archive (EMPIAR) with the accession codes EMPIAR-13138. Cryo-EM maps have been deposited in the Electron Microscopy Data Bank (EMDB) with accession codes EMD-50855 (CliM-SRC with P-tRNA and RF), EMD-50856 (CliM-SRC with P-tRNA and vacant A-site) and EMD-50858 (CliM-SRC with P- and A-tRNA). Molecular models have been deposited in the Protein Data Bank with accession code 9FY1 (CliM-SRC with P-tRNA and RF), 9FY2 [https://doi.org/10.2210/pdb9FY3/pdb] (CliM-SRC with P- tRNA and vacant A-site) and 9FY3 (CliM-SRC with P- and A-tRNA). Publicly available data used included PDB ID 9MTP, 8QOA, 5NWY, 7O19, 5A81, 8CVK, and 7NSO. Source data are provided with this paper.

## Code availability

MD simulation input files, final coordinates and analyses are available on zenodo.org (https://doi.org/10.5281/zenodo.17779011). Code for toeprinting visualization and deep mutational scanning analysis is available on GitHub at https://github.com/KigFjwr/toeprinteR/releases/tag/v0.0.2 and https://github.com/KigFjwr/arrest-peptide-DMS/releases/tag/v0.3.0, respectively.

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

## Acknowledgements

We thank Machiko Murata and Naoko Muraki for their technical support. This research was conducted within the Max Planck School Matter to Life, supported by the Dieter Schwarz Foundation and the German Federal Ministry of Research, Technology and Space (BMFTR) in collaboration with the Max Planck Society (O.B.). Part of this work was performed at the Multi-User CryoEM Facility at the Centre for Structural Systems Biology, Hamburg, supported by the Universität Hamburg and DFG grant numbers (INST 152/772-1|152/774-1|152/775-1|152/776-1|152/ 777-1 FUGG), the Federal Ministry of Education and Research (BMBF) and the DLR Projektträger (project SEEK 01KX2220). This work was supported by JSPS Grant-in-Aid for Scientific Research (Grant No. 20H05926, 21K06053, and 25K02230 to S.C., 19K16044, and 21K15020 to K.F., 23K05017 to H.T., 21K07018, and 25K01926 to N.O.), JST, PRESTO (JPMJPR24ND to K.F.), ACT-X (JPMJAX21BC to H.T.), Takeda Science Foundation (to S.C. and H.T.), Institute for Fermentation, Osaka (G-2021-2-063 to S.C., G-2024-2-071 to K.F., and G-2025-2-093 to H.T.), and by the Deutsche Forschungsgemeinschaft (DFG, German Research Foundation) WI3285/13-1 (to D.N.W.) and cluster of excellence Multiscale Bioimaging (EXC 2067/1-390729940 to L.V.B. and H.G.). We also acknowledge financial support from the Open Access Publication Fund of the Universität Hamburg.

## Author contributions

M.Y., K.F., and S.C. performed biochemical and genetic analyses. H.T. and N.O. prepared the Cd ribosome. H.A.S. prepared cryo-EM grids, screened and collected the cryo-EM data. F.G. performed the cryo-EM analysis, as well as generated and refined the molecular models, with help from H.P. MD simulations were performed by O.B., H.G. and L.V.B. S.C., K.F., D.N.W., and F.G. wrote the manuscript with input from all authors. S.C. and D.N.W. conceived and supervised the project.

## Funding

## Competing interests

The authors declare no competing interests.
