## [Transparent Peer Review File · Nature Communications]

Diverse mechanisms of translation arrest by a Clostridia ribosome stalling peptide CliM

Corresponding Author: Professor Daniel Wilson

Version 0:

Reviewer comments:

Reviewer #1

(Remarks to the Author)

The manuscript reports a comprehensive analysis, using structural, biochemical, and mutational approaches, of the stalling mechanisms of the CliM stalling peptide. CliM is a membrane-insertion-sensing arrest peptide that couples YidC activity to feedback regulation of yidC expression. YidC is a membrane protein insertase in bacteria that helps newly synthesized membrane proteins insert into, and fold within, the lipid bilayer. As I described below, the study is strong with almost every conclusion supported by two or more experimental approaches. I just have a few small comments and suggestions regarding experiments in Fig 1, toeprinting experiments in Fig 2, and the cryo-EM section. I am describing these concerns below in the context of the specific experiments.

1. The authors demonstrate that CliM is a bona fide YidC-monitoring arrest peptide that regulates downstream yidC2 expression in response to membrane insertase activity. Using reporter assays in *B. subtilis*, they show that inhibition of CliM membrane insertion enhances yidC2 expression in an arrest-dependent manner, while successful insertion releases arrest. mRNA secondary structure analysis reveals that a stem-loop downstream of cliM mediates this regulation by coupling ribosome stalling to yidC2 translation. In Figure 1, the authors validate the proposed regulatory model in which CliM functions as a YidC activity sensor. Using *spolIJ*⁺ (wild-type YidC1) and Δ *spolIJ* backgrounds as readouts of membrane-insertion capacity, the authors show that loss of *SpolIJ* selectively induces yidC2 expression in an arrest-dependent manner (Fig. 1b), while efficient insertion in the *spolIJ*⁺ background relieves CliM stalling (Fig. 1c). Arrest-deficient CliM mutants abolish yidC2 induction even when *SpolIJ* is absent, demonstrating that translational stalling, rather than *SpolIJ* deletion per se, drives the response. Dissection of the downstream mRNA stem-loop (Fig. 1d) further shows how ribosome stalling is transduced into increased yidC2 translation, completing the regulatory circuit. Together, these experiments convincingly establish that CliM couples YidC activity to feedback regulation of yidC2 expression. These are the main experiments in the paper to validate the YidC2 regulatory model.

While these results support the proposed model, I would have expected a bigger difference in beta-galactose activity for the wild type construct between *spolIJ*⁺ and Δ *spolIJ* in Fig 1b. Why does the wt construct in the *spolIJ*⁺ produce so much signal if, in principle, there should be efficient CliM insertion, consequent arrest release, and thus low yidC2 expression? Also, how do the authors know that the key arrest-deficient mutations are solely affecting arrest and they are not altering membrane targeting? Perhaps the authors could clarify these aspects of the essay.

2. Using toeprinting and in vitro translation assays, the authors demonstrate that *C. kluyveri* CliM stalls ribosomes at two adjacent codons during elongation. In contrast to the *C. kluyveri* homolog, *C. difficile* CliM stalls at a single site corresponding to a termination event, with a stop codon positioned in the A-site. Toeprinting and biochemical analyses confirm that ribosomes reach the stop codon but fail to undergo peptidyl-tRNA hydrolysis, indicating arrest during translation termination.

A concern with these assays is that the differences observed between species may be due to limitations in toeprinting experiments to unambiguously define the stall site, caused by reverse transcriptase pausing or by RNA adopting different structures. Is that a possibility given the way the analysis was performed in this study?

3. The authors used *C. difficile* CliM to create a stalled ribosome and to study it using cryo-EM and single-particle analysis. The fact that *C. difficile* CliM stalls at a single site with the Phe76 codon in the P-site and a stop codon in the A-site makes

this an ideal sample for this analysis, rather than using a ribosome stalled with the *C. kluyveri* CliM that stalls at multiple sites. Three ribosomal states were resolved. All of them contained the P-tRNA-Phe linked to the NC. The three classes contain either an A-tRNA (the smallest class) or an empty A site or a bound release factor in the A-site, confirming that arrest occurs after RF binding but prior to peptide release. It is proposed that the A-tRNA bound state contains tRNA^{Tyr}, which normally decodes UAC and UAU codons and has miscoded the UAA codon. Therefore, the most relevant class is the one bound to the RF and confirms that arrest occurs after RF binding but prior to peptide release. Structural analysis shows that CliM engages in extensive stacking, hydrogen-bonding, and hydrophobic interactions with 23S rRNA throughout the NPET. Many of these contacts involve residues identified as critical in the mutational scan, providing strong structure–function correlation. The authors demonstrate that CliM also makes specific interactions with the β -hairpin of uL22. Deletion of the uL22 tunnel loop abolishes arrest in vivo, whereas analogous deletions in uL4 or uL23 do not, establishing uL22 as a key structural determinant of CliM-mediated stalling. Comparison with accommodated release factor structures reveals that the penultimate CliM residue sterically blocks GGQ-loop accommodation at the PTC. Mutational analysis shows that reducing side-chain volume weakens or abolishes arrest, while larger residues preserve stalling, directly linking steric occlusion to arrest efficiency and stall-site position.

The structural analysis is excellent, and the cryo-EM maps are of high quality and resolution (~2.4 Å), specifically the density representing the nascent peptide. This is important because the mechanistic proposal of how the assembly is implemented requires the accurate positioning of key residues in the NC, and also nucleotides and r-proteins in the NPET. In particular, the claim that the position of the GGQ-loop in domain III of the RF, typically extending into the PTC, is displaced away by 3-4 Å. This is shown in Figure 7 with the molecular model, but the figure should also include a display of the relevant densities of the map for this region to support the accurate positioning of the model and ultimately the stated conclusion. Other than that, most of the conclusions derived from the structure are also validated with deletions and mutational assays, such as the involvement of uL22 in the stalling mechanism versus the requirement of, interactions with uL4 or uL23 that seem to be simply generic tunnel interactions.

Overall, I consider this manuscript very strong, and I support its publication after the authors address my concerns and comments described below.

Reviewer #2

(Remarks to the Author)

This manuscript presents a comprehensive and mechanistically insightful study of the arrest peptide CliM, establishing it as a functional ribosome stalling factor that senses YidC insertase activity. By integrating genetic reporter assays, in vitro translation, toeprinting, deep mutational scanning, high-resolution cryo-EM, and molecular dynamics simulations, the authors provide a coherent and largely convincing model explaining how CliM induces translational arrest during both elongation and termination. Overall, the study addresses a clearly defined and biologically relevant question, significantly expands the repertoire of known arrest peptides, and offers conceptual advances in our understanding of nascent chain-mediated translational control.

Major Concerns

1. The authors' main conclusion is that Leu75 prevents accommodation of either the GGQ motif of the release factor or the aminoacyl moiety of the A-site tRNA. The proposed rationale is that the helical structure of CliM constrains the position of Leu75, preventing it from undergoing the conformational rearrangement required for accommodation. However, a critical piece of evidence that is currently missing is a clear demonstration of how the CliM helix mechanically restricts the mobility of Leu75, especially given that Leu75 is already spatially distant from the CliM helix in the structure. Moreover, mutation of Leu75 to Gly results in +1 site stalling, indicating that this region retains a certain degree of conformational flexibility. Analogous to ErmDL-mediated arrest, which requires the additional presence of a macrolide antibiotic bound within the NPET to induce stalling, it remains unclear whether CliM-induced stalling at Leu75 is entirely dependent on the upstream helical elements. This issue could potentially be addressed by a more in-depth analysis integrating the deep mutational scanning data.

Minor Comments

1. It is recommended to include an additional table that integrates the deep mutational scanning data with structure-based interaction analysis, which would facilitate a clearer assessment of the consistency between the structural information and the mutational effects.
2. The conformation of the RF GGQ loop in Figure 7 is critical to the proposed mechanism, and the corresponding cryo-EM map for this region should be shown to support the authors' conclusions.

Version 1:

Reviewer comments:

Reviewer #1

(Remarks to the Author)

I appreciate the authors addressing my concerns about the original submission and providing a revised version with the necessary changes. Specifically, the additional sentences discussing the results in Figure 1b, which show the beta-galactosidase assay data, and the revision of Figure 7. I think this is now an outstanding paper with extremely solid data

supporting all the conclusions. I support the publication in Nature Communications in the present form.

Reviewer #2

(Remarks to the Author)

The authors have nicely revised their manuscript and I have no further comments and fully support the publication of this work.

The reviewers' comments are shown in **bold**, and our point-by-point responses are presented in *Italics*.

Reviewer's Comments:

Reviewer #1 (Remarks to the Author)

The manuscript reports a comprehensive analysis, using structural, biochemical, and mutational approaches, of the stalling mechanisms of the CliM stalling peptide. CliM is a membrane-insertion–sensing arrest peptide that couples YidC activity to feedback regulation of yidC expression. YidC is a membrane protein insertase in bacteria that helps newly synthesized membrane proteins insert into, and fold within, the lipid bilayer. As I described below, the study is strong with almost every conclusion supported by two or more experimental approaches. I just have a few small comments and suggestions regarding experiments in Fig 1, toeprinting experiments in Fig 2, and the cryo-EM section. I am describing these concerns below in the context of the specific experiments.

1. The authors demonstrate that CliM is a bona fide YidC-monitoring arrest peptide that regulates downstream yidC2 expression in response to membrane insertase activity. Using reporter assays in *B. subtilis*, they show that inhibition of CliM membrane insertion enhances yidC2 expression in an arrest-dependent manner, while successful insertion releases arrest. mRNA secondary structure analysis reveals that a stem–loop downstream of cliM mediates this regulation by coupling ribosome stalling to yidC2 translation. In Figure 1, the authors validate the proposed regulatory model in which CliM functions as a YidC activity sensor. Using spoIIIJ⁺ (wild-type YidC1) and ΔspoIIIJ backgrounds as readouts of membrane-insertion capacity, the authors show that loss of SpoIIIJ selectively induces yidC2 expression in an arrest-dependent manner (Fig. 1b), while efficient insertion in the spoIIIJ⁺ background relieves CliM stalling (Fig. 1c). Arrest-deficient CliM mutants abolish yidC2 induction even when SpoIIIJ is absent, demonstrating that translational stalling, rather than SpoIIIJ deletion per se, drives the response. Dissection of the downstream mRNA stem–loop (Fig. 1d) further shows how ribosome stalling is transduced into increased yidC2 translation, completing the regulatory circuit. Together, these experiments convincingly establish that CliM couples YidC activity to feedback regulation of yidC2 expression. These are the main experiments in the paper to validate the YidC2 regulatory model.

While these results support the proposed model, I would have expected a bigger difference in beta-galactose activity for the wild type construct between spoIIIJ⁺ and ΔspoIIIJ in Fig 1b. Why does the wt construct in the spoIIIJ⁺ produce so much signal if, in principle, there should be efficient CliM insertion, consequent arrest release, and thus low yidC2 expression? Also, how do the authors know that the key arrest-deficient mutations are solely affecting arrest and they are not altering membrane targeting? Perhaps the authors could clarify these aspects of the essay.

The relatively high basal expression of the downstream gene in the SpoIIIJ⁺ strain is likely attributable to incomplete release of CliM-mediated arrest, even in the presence of SpoIIIJ. This interpretation is supported by the marked reduction in downstream gene expression observed in the arrest-deficient (KWm) mutant (Fig. 1b), as well as by the lower b-

galactosidase activity of the WT variant relative to KWm in the Ck_cliM-lacZ arrest reporter under SpoIIIJ-proficient conditions (Fig. 1c).

We consider two possible explanations for the incomplete arrest release. First, CliM may not be efficiently inserted into the membrane by B. subtilis SpoIIIJ. Alternatively, although membrane insertion by SpoIIIJ may occur efficiently, it may not fully relieve translational arrest. In either case, these effects likely arise from heterologous expression of Clostridium CliM in B. subtilis.

Although the use of a heterologous expression system may partially obscure the native physiological behavior of CliM, we believe that the data are nevertheless sufficient to support our overall conclusions. These possibilities had already been discussed in the main text in relation to the results shown in Fig. 1c. We have now added a statement indicating that a similar interpretation also applies to the results shown in Fig. 1b (as underlined below).

*“The lower activity of the WT reporter compared with the arrest-deficient mutant suggests that partial arrest persists even with the TM region, likely because B. subtilis SpoIIIJ does not efficiently insert heterologously expressed Ck CliM or fails to fully release arrest upon membrane insertion. **This notion may also account for the higher yidC'-lacZ induction observed for the WT reporter relative to the arrest-deficient mutant (Fig. 1b).**”*

Because Figs. 1b and 1c provide complementary data, we believe that this revision improves the balance and clarity of the presentation.

We cannot formally exclude the possibility that the arrest-deficient mutations affect membrane targeting. However, our conclusion that arrest is essential for downstream gene expression is supported by experiments using TM-deleted CliM variants, in which membrane targeting is inherently impaired and thus potential effects of the arrest-deficient mutations on membrane targeting do not need to be considered. Under these conditions, the arrest-deficient mutation markedly reduced induction of the downstream gene (Fig. 1b; DTM vs DTM/KWm), supporting sufficiently our conclusion that arrest plays an essential role in downstream gene induction.

2. Using toeprinting and in vitro translation assays, the authors demonstrate that C. kluveri CliM stalls ribosomes at two adjacent codons during elongation. In contrast to the C. kluveri homolog, C. difficile CliM stalls at a single site corresponding to a termination event, with a stop codon positioned in the A-site. Toeprinting and biochemical analyses confirm that ribosomes reach the stop codon but fail to undergo peptidyl-tRNA hydrolysis, indicating arrest during translation termination.

A concern with these assays is that the differences observed between species may be due to limitations in toeprinting experiments to unambiguously define the stall site, caused by reverse transcriptase pausing or by RNA adopting different structures. Is that a possibility given the way the analysis was performed in this study?

Although reverse transcription can be prematurely terminated due to RNA secondary structures, such signals are also observed in toeprint assays performed with arrest-deficient mutants (KWm) or in the presence of the translation inhibitor chloramphenicol. By including these negative controls, we are able to distinguish translation arrest-dependent toeprint

signals from reverse transcription stops caused by RNA structure. Our data were interpreted with these controls. Therefore, it is unlikely that the differences in stalling sites observed between the two CliM homologs arise from the RNA secondary structure.

3. The authors used *C. difficile* CliM to create a stalled ribosome and to study it using cryo-EM and single-particle analysis. The fact that *C. difficile* CliM stalls at a single site with the Phe76 codon in the P-site and a stop codon in the A-site makes this an ideal sample for this analysis, rather than using a ribosome stalled with the *C. kluyveri* CliM that stalls at multiple sites. Three ribosomal states were resolved. All of them contained the P-tRNA-Phe linked to the NC. The three classes contain either an A-tRNA (the smallest class) or an empty A site or a bound release factor in the A-site, confirming that arrest occurs after RF binding but prior to peptide release. It is proposed that the A-tRNA bound state contains tRNA^{Tyr}, which normally decodes UAC and UAU codons and has miscoded the UAA codon. Therefore, the most relevant class is the one bound to the RF and confirms that arrest occurs after RF binding but prior to peptide release. Structural analysis shows that CliM engages in extensive stacking, hydrogen-bonding, and hydrophobic interactions with 23S rRNA throughout the NPET. Many of these contacts involve residues identified as critical in the mutational scan, providing strong structure–function correlation. The authors demonstrate that CliM also makes specific interactions with the β -hairpin of uL22. Deletion of the uL22 tunnel loop abolishes arrest *in vivo*, whereas analogous deletions in uL4 or uL23 do not, establishing uL22 as a key structural determinant of CliM-mediated stalling. Comparison with accommodated release factor structures reveals that the penultimate CliM residue sterically blocks GGQ-loop accommodation at the PTC. Mutational analysis shows that reducing side-chain volume weakens or abolishes arrest, while larger residues preserve stalling, directly linking steric occlusion to arrest efficiency and stall-site position.

The structural analysis is excellent, and the cryo-EM maps are of high quality and resolution (~2.4 Å), specifically the density representing the nascent peptide. This is important because the mechanistic proposal of how the assembly is implemented requires the accurate positioning of key residues in the NC, and also nucleotides and r-proteins in the NPET. In particular, the claim that the position of the GGQ-loop in domain III of the RF, typically extending into the PTC, is displaced away by 3-4 Å. This is shown in Figure 7 with the molecular model, but the figure should also include a display of the relevant densities of the map for this region to support the accurate positioning of the model and ultimately the stated conclusion.

The density for the GGQ loop of the RF is shown in ED Fig 2, however, we have now revised Figure 7 panel b to also show the density for the GGQ loop there too as requested.

Other than that, most of the conclusions derived from the structure are also validated with deletions and mutational assays, such as the involvement of uL22 in the stalling mechanism versus the requirement of, interactions with uL4 or uL23 that seem to be simply generic tunnel interactions. Overall, I consider this manuscript very strong, and I support its publication after the authors address my concerns and comments described below.

Reviewer #2 (Remarks to the Author)

This manuscript presents a comprehensive and mechanistically insightful study of the arrest peptide CliM, establishing it as a functional ribosome stalling factor that senses YidC insertase activity. By integrating genetic reporter assays, *in vitro* translation, toeprinting, deep mutational scanning, high-resolution cryo-EM, and molecular dynamics simulations, the authors provide a coherent and largely convincing model explaining how CliM induces translational arrest during both elongation and termination. Overall, the study addresses a clearly defined and biologically relevant question, significantly expands the repertoire of known arrest peptides, and offers conceptual advances in our understanding of nascent chain-mediated translational control.

Major Concerns

1. The authors' main conclusion is that Leu75 prevents accommodation of either the GGQ motif of the release factor or the aminoacyl moiety of the A-site tRNA. The proposed rationale is that the helical structure of CliM constrains the position of Leu75, preventing it from undergoing the conformational rearrangement required for accommodation. However, a critical piece of evidence that is currently missing is a clear demonstration of how the CliM helix mechanically restricts the mobility of Leu75, especially given that Leu75 is already spatially distant from the CliM helix in the structure. Moreover, mutation of Leu75 to Gly results in +1 site stalling, indicating that this region retains a certain degree of conformational flexibility. Analogous to ErmDL-mediated arrest, which requires the additional presence of a macrolide antibiotic bound

within the NPET to induce stalling, it remains unclear whether CliM-induced stalling at Leu75 is entirely dependent on the upstream helical elements. This issue could potentially be addressed by a more in-depth analysis integrating the deep mutational scanning data.

The basis for the proposal that the helix mechanically restricts the mobility of Leu75 comes primarily from the molecular dynamics simulations that show that when the helix is in place, there is relatively little mobility of Leu and all accessible conformations would overlap with the RF (Fig. 8a, WT), however, once the helix is unwound the nascent chain naturally has more degrees of freedom. This hypothesis is supported by the mutagenesis data showing that proline substitutions that disrupt helix formation also lead to relief of stalling, which necessitates the movement of Leu75 out of the A-site pocket.

The suggestion from the reviewer that the mutation of Leu75 to Gly leads to stalling in the +1 position shows that the region retains a certain degree of flexibility, may have arisen from a misunderstanding since the Leu75 to Gly mutation only leads to +1 stalling when the stop codon is mutated to lysine (K77), as was used in the DMS analysis, and therefore does not occur in the natural Cd CliM staller that was used for the cryo-EM analysis. Moreover, even if we consider that stalling at the +1 position does lead to incorporation of another amino acid into the nascent chain, we do not know whether the mechanism of stalling at the +1 site is similar to that proposed here for CliM i.e. whether helix formation is actually required for stalling at the +1 site. Nevertheless, we cannot rule out that the PTC retains some plasticity such that the additional incorporation of another amino acid is possible, however, further work would be needed to address this hypothesis.

Minor Comments

1. It is recommended to include an additional table that integrates the deep mutational scanning data with structure-based interaction analysis, which would facilitate a clearer assessment of the consistency between the structural information and the mutational effects.

We have now added a new Supplementary Fig. 4 that compares the deep mutational scanning data directly with interaction analysis from the cryo-EM structure of the CliM-SRC.

2. The conformation of the RF GGQ loop in Figure 7 is critical to the proposed mechanism, and the corresponding cryo-EM map for this region should be shown to support the authors' conclusions.

The density for the GGQ loop of the RF is shown in ED Fig 2, however, we have now revised figure 7b to also show the density for the GGQ loop there too, as also requested by Reviewer #1 (see response above with inserted revised Figure 7).